# Warming-induced vapor pressure deficit suppression of vegetation growth diminished in northern peatlands

Ning Chen[1,2], Yifei Zhang[1], Fenghui Yuan[1,3], Changchun Song[1,4] ✉, Mingjie Xu[5], Qingwei Wang [2], Guangyou Hao[2], Tao Bao[6], Yunjiang Zuo[1], Jianzhao Liu[1,7], Tao Zhang[5], Yanyu Song[1], Li Sun[1], Yuedong Guo[1], Hao Zhang[1], Guobao Ma[1], Yu Du[1], Xiaofeng Xu [8] ✉ & Xianwei Wang[1] ✉

Recent studies have reported worldwide vegetation suppression in response to increasing atmospheric vapor pressure deficit (VPD). Here, we integrate multisource datasets to show that increasing VPD caused by warming alone does not suppress vegetation growth in northern peatlands. A site-level manipulation experiment and a multiple-site synthesis find a neutral impact of rising VPD on vegetation growth; regional analysis manifests a strong declining gradient of VPD suppression impacts from sparsely distributed peatland to densely distributed peatland. The major mechanism adopted by plants in response to rising VPD is the "open" water-use strategy, where stomatal regulation is relaxed to maximize carbon uptake. These unique surface characteristics evolve in the wet soil–air environment in the northern peatlands. The neutral VPD impacts observed in northern peatlands contrast with the vegetation suppression reported in global nonpeatland areas under rising VPD caused by concurrent warming and decreasing relative humidity, suggesting model improvement for representing VPD impacts in northern peatlands remains necessary.

Vegetation dynamics are critical in shaping carbon balances and generating biophysical feedback within the climate system by modifying surface albedo and the energy budget[1–5]. Vegetation greening was evident before the late 1990s and has now stalled or reversed because of a sharp increase in global vapor pressure deficit (VPD), partially offsetting positive $CO_2$ fertilization effects[6]. These observations indicate that vegetation greening is usually negatively correlated with increasing VPD[6–10]. The prolonged period of high VPD has been acknowledged as a major driver of large-scale tree mortality[11,12], global vegetation greening reversal[7,8,13], and carbon sink reduction[13]. These negative impacts usually occur with increasing VPD ($VPD = f(Ta, RH)$) caused by warming air temperature (Ta) and declining relative humidity (RH) or are covaried[6,14]. Although VPD shows a continuously increasing trend after the 1990s[6,15], its effects on vegetation and the underlying mechanisms may vary among regions due to different environmental conditions[16].

Although the typically negative VPD effects have been widely discussed recently, the increasing VPD may be driven by different

[1]Key Laboratory of Wetland Ecology and Environment, Northeast Institute of Geography and Agroecology, Chinese Academy of Sciences, 130102 Changchun, China. [2]CAS Key Laboratory of Forest Ecology and Management, Institute of Applied Ecology, Chinese Academy of Sciences, 110016 Shenyang, China. [3]Department of Soil, Water, and Climate, University of Minnesota, Saint Paul, MN 55108, USA. [4]School of Hydraulic Engineering, Dalian University of Technology, 116024 Dalian, China. [5]College of Agronomy, Shenyang Agricultural University, 110866 Shenyang, China. [6]Key Laboratory of Regional Climate-Environment for Temperate East Asia, Institute of Atmospheric Physics, Chinese Academy of Sciences, Beijing, China. [7]College of Surveying and Exploration Engineering, Jilin Jianzhu University, 130018 Changchun, China. [8]Biology Department, San Diego State University, San Diego 92182, USA. ✉e-mail: songcc@iga.ac.cn; xxu@sdsu.edu; wangxianwei@iga.ac.cn

mechanisms. A recent global study has detected a subtle increase in RH in northern peatlands over the last four decades[17] (Supplementary Fig. 1), suggesting that the actual water vapor (AVP) increases at approximately the same rate as the saturation water vapor (SVP) ($RH = f(AVP, SVP)$). This may be due to the unique surface characteristics of the water-rich environment and the high moss cover in the northern peatlands, which supplies sufficient atmospheric water to meet the increasing water demand caused by high VPD[18] and to maintain the atmospheric water balance. If RH remains unchanged, warming alone may lead to widespread increases in atmospheric VPD[19]. Over the same period of subtle changes in RH and substantial increases in VPD[6], a neutral response of vegetation growth to VPD has been detected from recent analyses[6,13,20,21]. Multiple lines of evidence have raised the concern that vegetation growth may respond differently to increasing VPD caused by warming alone, contrasting to that caused by concurrent warming and decreased RH. However, no direct observational evidence has been reported for whether increasing VPD can be driven by warming alone or how vegetation growth responds to warming-induced increases in VPD[16].

The main mechanism behind the VPD effects in the northern peatlands may be different from those in other ecosystems. Increasing atmospheric demand for water induced by rising VPD suppresses photosynthesis and transpiration of plants by controlling stomatal activity and xylem conductance; therefore, increasing VPD plays a critical role in regulating the water and carbon cycles of terrestrial ecosystems[10,16,22]. VPD can also indicate atmospheric dryness. A typical plant response to increasing VPD is stomatal closure to minimize water loss and avoid excessive water tension within the xylem at the expense of reducing or stopping photosynthesis[7,10,22–24]. It is well known that the vegetation response to increasing VPD is always negative[6–10,22,25]. In contrast, photosynthesis in the Amazon rainforest increases as the VPD increases[26], mainly due to new leaves flushed during the dry season compensating for the stomatal limitations caused by increased atmospheric dryness[25]. A moist and cold environment combined with high moss cover may produce sufficient atmospheric moisture to meet the increasing water demand caused by increasing VPD in northern peatlands[18], potentially relieving atmospheric dryness[27,28]. In this wet soil–air environment, plants may adopt an "open" water-use strategy in response to water stress by relaxing stomatal regulation to maximize carbon uptake at the cost of hydraulic risk[29].

Here, we compiled multisource datasets of in situ observations, a multisite synthesis (78 sites), eddy covariance flux towers (18 sites from FLUXNET-CH$_4$ Community Product and 95 sites from FLUX-NET2015), and regional-scale remote sensing products to explore the effects of increasing VPD and investigate their underlying mechanisms in northern peatlands. We further compared the northern peatlands with global nonpeatlands to identify the VPD impacts on vegetation in northern peatlands that differ from the prevailing viewpoint of VPD suppression on vegetation. Our data implied that the suppressive effect of rising VPD driven by co-occurring warming and decreasing RH on vegetation growth in the global nonpeatland regions was greater than that caused by warming alone in the northern peatlands. Differences in surface characteristics (e.g., water availability) and plant traits (e.g., plant water-use strategy) were the major factors explaining these contrasting VPD effects.

## Results and discussion
### Vegetation responses to warming-induced rising VPD in northern peatlands
Synthesized warming experiments showed no suppressive impact of rising VPD on vegetation growth in the northern peatlands. Sixty-seven synthesized warming experiments with observed Ta and vegetation growth and simulated VPD from the CRU 4.04 datasets were extracted from a recent warming meta-analysis across the northern peatlands[30]. The CRU 4.04 datasets of Ta and VPD matched well with the

observations in 18 eddy covariance towers, with significant correlation coefficients ($r$) of $0.98 \pm 0.03$ (mean $\pm$ 1 standard error, se) and $0.93 \pm 0.02$ for Ta and VPD, respectively ($p < 0.05$, Supplementary Fig. 2a, b, Methods). Eddy covariance flux data showed that VPD exponentially increased with Ta ($p < 0.05$, mean $\pm$ 1 se, $r = 0.93 \pm 0.01$, Supplementary Fig. 2c, Methods). We then fitted the exponential relationships between Ta and VPD from the CRU 4.04 datasets ($p < 0.05$, mean $\pm$ 1 se, $r = 0.91 \pm 0.004$) to simulate VPD differences between the warming and control treatments at 67 sites (Supplementary Table 1, Methods). Regression analyses showed that the warming impacts on net primary productivity (NPP) had no relationship with VPD across sites, suggesting that vegetation in the northern peatlands was not affected by rising VPD (regression coefficients $\pm$ 1 se, $-0.02 \pm 0.08$, $p = 0.769$, Fig. 1a).

Among the observations of VPD and vegetation growth in warming experiments, we found that rising VPD caused by warming alone did not significantly reduce vegetation growth in the northern peatlands. Observational data from a warming experiment in Mohe showed that the warming treatment elevated Ta by $3.8 \pm 0.6$ °C (mean $\pm$ 1 se, $p < 0.001$) and decreased RH by $1.1 \pm 0.6\%$ ($p = 0.096$), leading to a significant increase in VPD of $1.5 \pm 0.3$ hPa ($p < 0.001$, Fig. 1b–d). However, vegetation growth (measured by canopy conductance, $G_c$, of *Vaccinium uliginosum*) did not show suppression in response to the increasing VPD caused by warming alone (regression coefficients $\pm$ 1 se, $0.23 \pm 0.48$, $p = 0.642$, Fig. 1e).

To verify these findings at the Mohe site, we synthesized six other field warming experiments from 273 independent sites in a meta-analysis[30]. In these warming experiments, Ta, VPD (or RH), and an indicator of vegetation growth, such as biomass, were included in the warming and control treatments (Supplementary Table 2, Methods). Compared with the control treatment, the warming treatments resulted in significant increases in Ta ($p = 0.002$) and insignificant reductions in RH ($p = 0.083$), with values of $2.7 \pm 0.6$ °C (mean $\pm$ 1 se) and $8.0 \pm 4.0\%$, respectively, leading to a significant increase in VPD of $2.6 \pm 1.1$ hPa ($p = 0.043$, Fig. 1f–h). The synthesized results confirmed that the growing VPD induced by warming alone did not yield significant suppression impacts on vegetation growth in the northern peatlands (regression coefficients $\pm$ 1 se, $-0.11 \pm 0.27$, $p = 0.701$, Fig. 1i).

We extended the analysis to the northern peatland area to verify the neutral VPD impacts on vegetation activities at a larger scale. Across the entire northern peatlands, Ta (mean $\pm$ 1 se, $0.028 \pm 0.006$ °C yr$^{-1}$, $p < 0.001$) and VPD ($0.004 \pm 0.002$ hPa yr$^{-1}$, $p = 0.039$) substantially increased from 1982 to 2018, while RH ($0.018 \pm 0.010\%$ yr$^{-1}$, $p = 0.076$) remained unchanged (Fig. 2a–c). Over the 37 years, more than 60.0% of the regions with significant increases in VPD had significant increases in Ta ($p < 0.05$) but not in RH ($p > 0.05$) over the northern peatlands (Supplementary Fig. 3). Partial correlation (PCOR) analyses showed that the regional mean PCOR coefficients (PCOR$_{GPP \text{ vs. } VPD}$) of the three datasets of satellite-derived gross primary productivity (GPP) were 0.06–0.08 (Fig. 2d–f). Three detrended satellite-derived GPP positively correlated with detrended VPD over 58.2 to 66.4% (24.8 to 33.0% with a significant positive correlation at $p = 0.05$) of the northern peatlands when detrended Ta, radiation, wind speed, and precipitation were considered (Fig. 2d–f, Methods). In contrast, a significant negative response of GPP to VPD was found in only 8.1–16.8% of the northern peatlands ($p < 0.05$, Fig. 2d–f). The suppression impact weakened with increasing peatland extent. As the peatland extent increased from 10% to >70%, the regional mean PCOR$_{GPP \text{ vs. } VPD}$ increased from $0.09 \pm 0.01$ (mean $\pm$ 1 se for three satellite-derived GPP) to $0.17 \pm 0.02$, and the percentage of significantly negative PCOR$_{GPP \text{ vs. } VPD}$ decreased from $8.58 \pm 2.35$ to $0.81 \pm 0.44$ (Fig. 2g–l). This further suggested that increasing VPD had a neutral impact on vegetation growth in the northern peatlands, especially for the densely distributed peatlands.

## Vegetation responses to rising VPD caused by concurrent warming and decreasing RH in global nonpeatland regions

Increasing VPD caused by concurrent warming and decreasing RH led to a stronger suppression impact on GPP in global nonpeatland regions. Ta (mean ± 1 se, $0.027 \pm 0.003 \,°C \, yr^{-1}$, $p < 0.001$) and VPD ($0.018 \pm 0.001 \, hPa \, yr^{-1}$, $p < 0.001$) increased significantly during 1982−2018, but RH ($-0.027 \pm 0.005\% \, yr^{-1}$, $p < 0.001$) decreased

significantly in global nonpeatland regions (Fig. 3a–c). From 1982 to 2018, ~60% of the global nonpeatland regions showed an increasing VPD trend with significantly increasing Ta and significantly declining RH ($p < 0.05$, Supplementary Fig. 4). When the detrended Ta, radiation, wind speed, and precipitation were considered, PCOR analyses showed that three detrended satellite-derived GPP were negatively correlated with the detrended VPD over 65.7–72.7% of the global

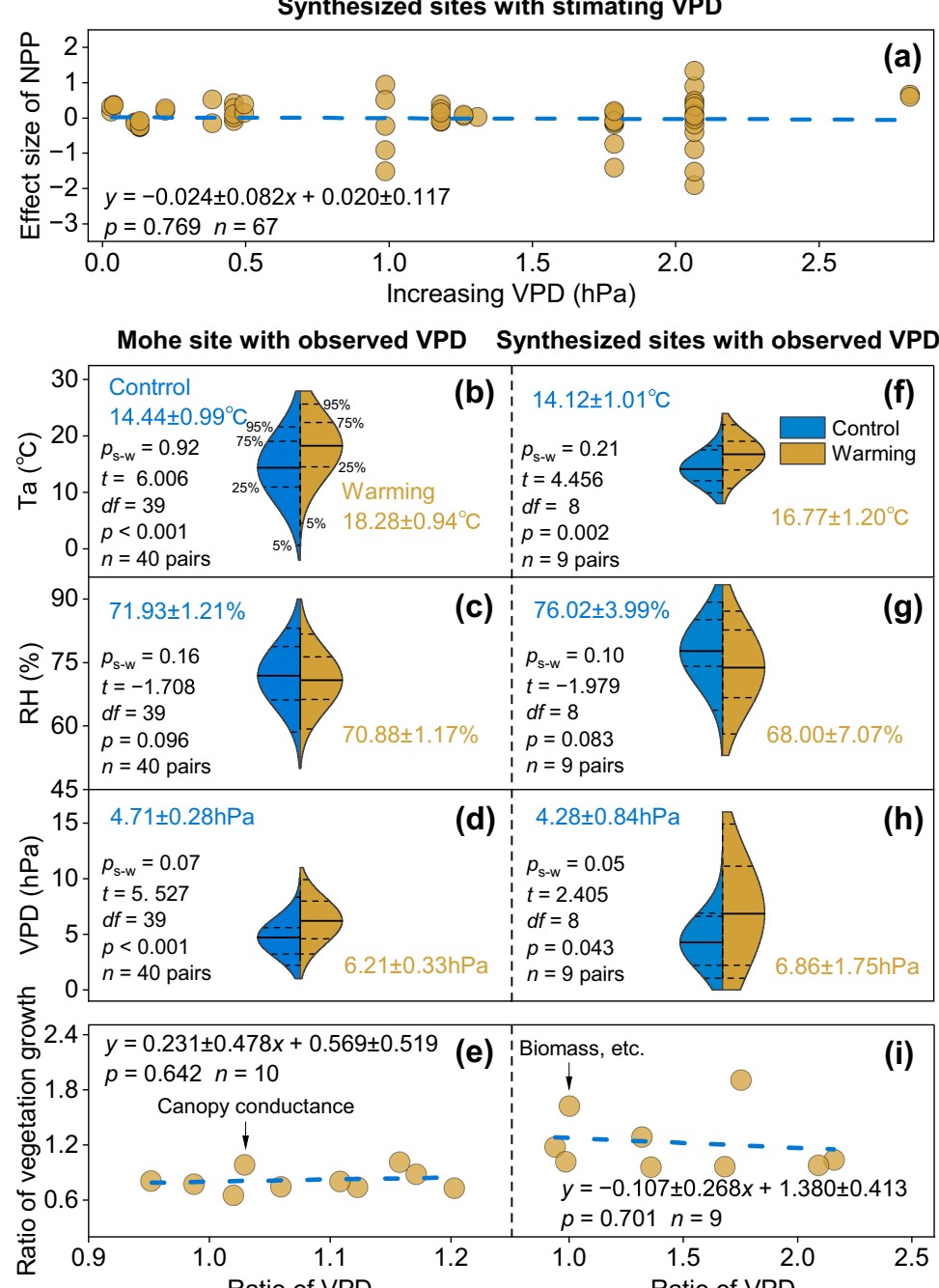

**Fig. 1 | Changes in air temperature (Ta), relative humidity (RH), and vapor pressure deficit (VPD) and VPD impacts on vegetation growth. a**, **e** and **i** VPD impacts on vegetation growth at 67 warming experiment sites stimulating VPD (**a**), Mohe site (**e**), and synthesized warming experiment sites (**i**) with observed VPD under the control and warming treatments. Vegetation growth is quantified as canopy conductance of *Vaccinium uliginosum* at the Mohe site (**e**) and by biomass (**i**), etc., at the synthesized sites. Ratio: ratio of vegetation growth and VPD in the warming treatment to the control treatment (**e** and **i**). *n* represents the total

number of observations (**a**–**i**). **b** and **f**, **c** and **g**, **d** and **h** Changes in Ta, RH, and VPD under control (blue) and warming (brown) treatments, respectively. Black lines indicate the means of Ta, RH, and VPD under warming and control treatments in the violin plots. Dotted lines indicate 5%, 25%, 75%, and 95% percentiles in the violin plots. $p_{s-w} > 0.05$ indicates that differences in Ta, RH, and VPD between warming and control treatments meet the Shapiro–Wilk test for normality. *p* values, *t*-statistic (*t*), and degrees of freedom (*df*) are the result of *t*-tests of Ta, RH, and VPD between control and warming treatments.

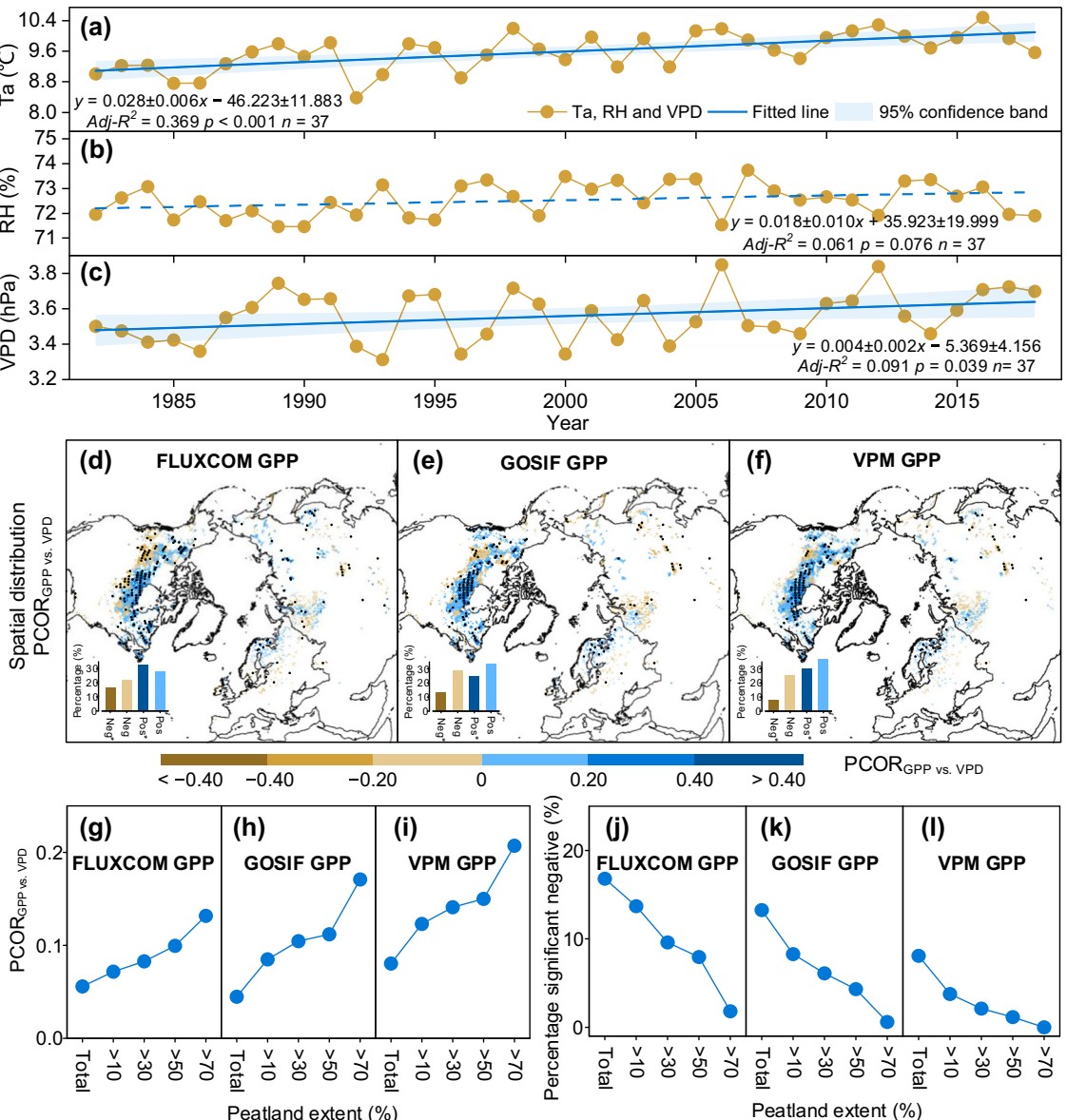

**Fig. 2 | Changing rates of air temperature (Ta), relative humidity (RH), vapor pressure deficit (VPD), and spatial distributions of the VPD impacts on vegetation growth across the northern peatlands during 1982–2018. a–c** Changing rates of Ta, RH, and VPD from 1982 to 2018. Solid and dotted lines indicate the significant and insignificant variability of Ta, RH, and VPD, respectively. Shading areas along the regression lines are the 95% confidence band. *Adj-R²*: Adjusted R Square. *n* represents the total number of observations. **d–f** The spatial distributions of the $PCOR_{GPP\ vs.\ VPD}$ derived from FLUXCOM GPP, GOSIF GPP, and VPM GPP. Insets show the percentage (%) of significant (Neg*) and insignificant (Neg) negative (brown) $PCOR_{GPP\ vs.\ VPD}$ and significant (Pos*) and insignificant (Pos) positive (blue) $PCOR_{GPP\ vs.\ VPD}$. **g–l** Changes in regional mean $PCOR_{GPP\ vs.\ VPD}$ (**g–i**) and the percentage of significant negative $PCOR_{GPP\ vs.\ VPD}$ (**j–l**) across the peatland extent.

nonpeatland regions (44.3–58.2% with a significant negative correlation, $p < 0.05$) (Fig. 3d, Supplementary Fig. 5). The regional mean $PCOR_{GPP\ vs.\ VPD}$ ranged from −0.16 to −0.21 for the three satellite-derived GPP (Fig. 3d). The spatial coverage of significantly negative $PCOR_{GPP\ vs.\ VPD}$ was 40% higher and the regional mean $PCOR_{GPP\ vs.\ VPD}$ was 0.25 lower in the global nonpeatland regions than those in the northern peatlands (Fig. 3d).

To further assess the robustness of the divergent drivers and impacts of increasing VPD, the global nonpeatland regions were divided into nonhumid regions (AI < 0.65) and humid regions (AI ≥ 0.65) based on the aridity index (AI, Methods)[31]. From 1982 to 2018, changes in the temporal trends of Ta, RH, and VPD in the nonhumid regions and the humid regions were consistent with those in the entire nonpeatland region (Supplementary Figs. 4, 6). The PCOR analyses, when the detrended Ta, radiation, wind speed, and precipitation were considered, showed that the spatial coverage of significantly negative $PCOR_{GPP\ vs.\ VPD}$ were 59.5% higher (mean of the three satellite-derived GPP) in the nonhumid regions and 25.4% greater in the humid regions, respectively ($p < 0.05$, Supplementary Figs. 5, 6), than those in the northern peatlands. In addition, a lower regional mean $PCOR_{GPP\ vs.\ VPD}$ from three satellite-derived GPP was observed in the nonhumid regions (−0.32) and the humid regions (−0.14) compared to the northern peatlands (0.06) (Supplementary Figs. 5, 6).

The VPD effects based on satellite-derived GPP in the northern peatlands and the global nonpeatland regions were validated by comparison against results with 113 eddy covariance flux towers. Observational data showed that detrended VPD was significantly negatively correlated with detrended GPP at 1 out of 18 (5.6%) eddy covariance flux towers in the northern peatlands and 43 out of 95 (45.3%) towers in global nonpeatland regions, respectively ($p < 0.05$,

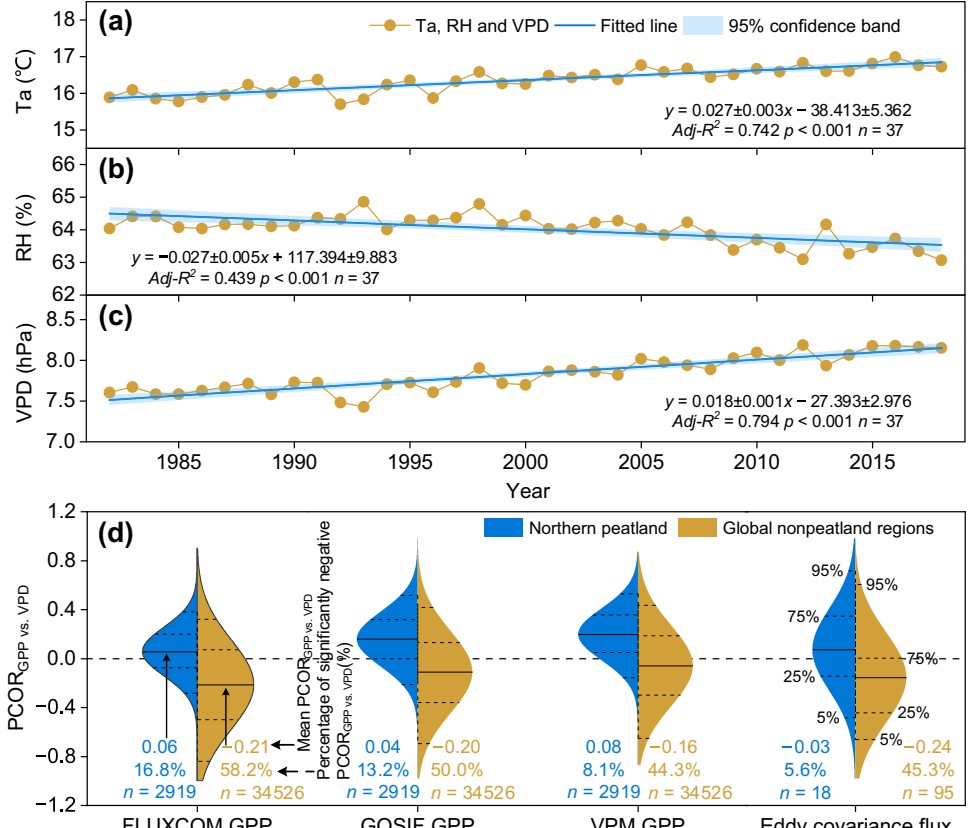

**Fig. 3 | Changing rates of air temperature (Ta), relative humidity (RH), and vapor pressure deficit (VPD) and the impact of VPD on vegetation growth in global nonpeatland regions. a–c** Changing rates of Ta, RH, and VPD from 1982 to 2018 in global nonpeatland regions. Shading areas along the regression lines are the 95% confidence band. *Adj-R²*: Adjusted R Square. **d** Differences in VPD effects as estimated by the mean PCOR$_{GPP vs. VPD}$ (horizontal solid arrow direction) and the percentage (%) of significantly negative PCOR$_{GPP vs. VPD}$ (horizontal dotted arrow direction) between the northern peatlands (blue) and the global nonpeatland regions (brown). Solid lines in the violin plots indicate the mean PCOR$_{GPP vs. VPD}$. Dotted lines indicate 5%, 25%, 75%, and 95% percentiles in the violin plots. *n* indicates the total number of observations.

Fig. 3d). Grouping the 95 eddy covariance flux towers into the non-humid regions (33 towers) and the humid regions (62 towers) further confirmed a greater VPD suppression impact in the global nonpeatland regions compared to the northern peatlands, with a significantly negative PCOR$_{GPP vs. VPD}$ of 63.6% and 35.5% in the nonhumid regions and the humid regions, respectively (Supplementary Fig. 6). In addition, according to the latitude, longitude, and time span in 113 eddy covariance flux towers, we found that the symbols (±) of the satellite-derived PCOR$_{GPP vs. VPD}$ agreed with 76.4% (mean value from three satellite-derived GPP) of the eddy covariance flux towers (Supplementary Fig. 7, Methods). In addition, the satellite-derived PCOR$_{GPP vs. VPD}$ was positively correlated with eddy covariance PCOR$_{GPP vs. VPD}$, with *r* values ranging from 0.51 to 0.58 ($p < 0.05$, Supplementary Fig. 7). In summary, the field-scale and grid-scale observations consistently suggested that the prevailing viewpoint derived from the global nonpeatland regions may overestimate the VPD suppression impact in the northern peatlands.

## Mechanisms for the divergent VPD effects

Six plant traits and environmental factors were used to understand the causes of the contrasting VPD effects. Soil hydraulic properties were measured by soil organic carbon (SOC) and bulk density (BD)[32]. Generally, higher SOC and lower BD indicated good soil hydraulic properties[32,33]. Climate water deficit (CWD) and available volumetric water content (VWC) were used to evaluate water availability[34–36]. A low CWD and a high VWC correspond to a wet environment. Plant water-use strategy was quantified by underlying water use efficiency (uWUE)

and transpiration (Et) response to VPD (PCOR$_{Et vs. VPD}$)[37]. The uWUE is inversely proportional to the marginal water cost of carbon gain (λ)[38,39] and is widely used as the proxy for plant water-use strategy[40–42]. For example, a high uWUE corresponds to a low λ, as a "reduce expenditures" water-use strategy, suggesting that plants tended to close stomata sooner and minimize water loss under drying conditions at the cost of carbon uptake[37,42]. Overall, in regions where a "reduce expenditure" and an "open" water-use strategy is more prevalent, this is associated with a higher and lower uWUE, respectively[40,41]. PCOR$_{Et vs. VPD}$ provided a direct proxy for the response of stomatal activity to increasing VPD, with a more positive value indicating weaker stomatal limitation and a more negative value indicating stronger stomatal limitation. In addition, PCOR$_{Et vs. VPD}$ can also be used as a proxy for atmospheric water supply in response to increasing water demand caused by increasing VPD[27].

Differing plant water-use strategies were a major reason for the contrasting VPD effects. We used the random forest algorithm - a machine learning approach - to relate the PCOR$_{GPP vs. VPD}$ (a proxy for VPD effects) as a function of six plant traits and environmental factors (Methods). The predicted PCOR$_{GPP vs. VPD}$ from the random forest models agreed well with the observed PCOR$_{GPP vs. VPD}$ ($r = 0.84$ to 0.85, $p < 0.001$, Fig. 4a). The sensitivity of PCOR$_{GPP vs. VPD}$ to changes in uWUE (mean ± 1 se, −0.80 ± 0.13) and PCOR$_{Et vs. VPD}$ (1.12 ± 0.19) was greater than that of the other variables (−0.71 ± 0.05 to 0.27 ± 0.07) (Fig. 4b, Methods). PCOR$_{GPP vs. VPD}$ decreased with increasing uWUE, but it increased with increasing PCOR$_{Et vs. VPD}$ (Fig. 4b). In the northern peatlands, the uWUE and the PCOR$_{Et vs. VPD}$ were 0.61 g C kPa⁰·⁵ kg⁻¹

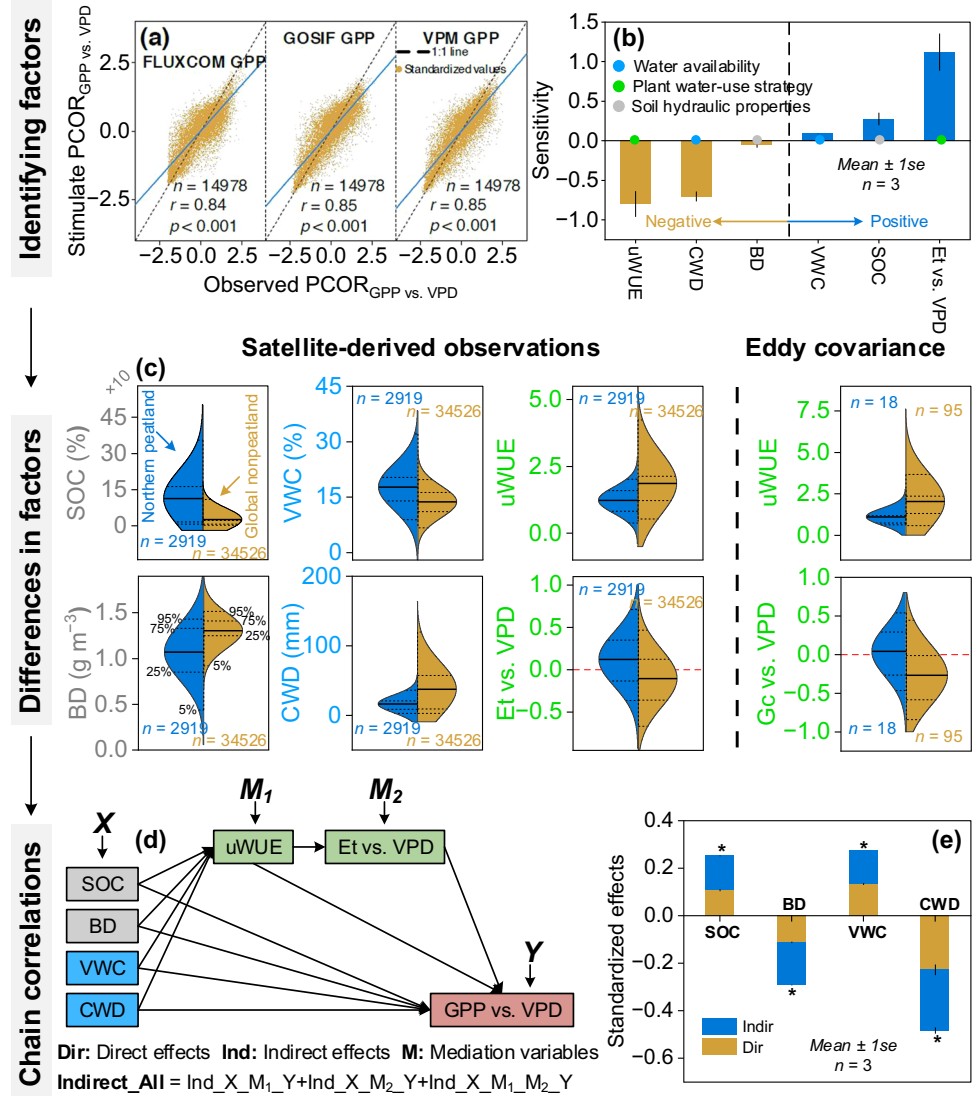

**Fig. 4 | Mechanisms for the divergent VPD effects between the northern peatlands and global nonpeatland regions. a** Correlations of observed $PCOR_{GPP \text{ vs. VPD}}$ from three satellite-derived GPP datasets with simulated $PCOR_{GPP \text{ vs. VPD}}$. **b** Sensitivities (mean ± 1 standard error; positive: blue histograms; negative: brown histograms) of the $PCOR_{GPP \text{ vs. VPD}}$ to water availability (VWC and CWD; blue dots), soil hydraulic properties (SOC and BD; gray dots), and plant water-use strategy ($PCOR_{Et \text{ vs. VPD}}$ and uWUE; green dots). **c** Differences in water availability, soil hydraulic properties, and plant water-use strategy between the northern peatlands and the global nonpeatland regions, including satellite-derived observations (left of dotted line) and eddy covariance flux towers (right of dotted line). Solid lines in the violin plots indicate the mean of these variables. Dotted lines indicate 5%, 25%, 75%, and 95% percentiles in the violin plots. $n$ indicates the total number of observations. **d** Schematic diagram illustrating the mediation effect models. Ind_X_M₁_Y, Ind_X_M₂_Y, and Ind_X_M₁_M₂_Y indicate that water availability ($X$) and soil hydraulic properties ($X$) influence the VPD effects ($Y$, GPP vs. VPD) through uWUE (M₁, Ind_X_M₁_Y), $PCOR_{Et \text{ vs. VPD}}$ (Et vs. VPD; M₂, Ind_X_M₂_Y), and both (Ind_X_M₁_M₂_Y). **e** Standardized direct and indirect effects (mean ± 1 standard error) of water availability and soil hydraulic properties on the VPD effects. "*" denotes that all the standardized effects are significant ($p < 0.05$). Error bars represent one unit of standard error ($n = 3$) (**b** and **e**).

H₂O lower and 0.22 higher than the global nonpeatland regions, respectively (mean ± 1 se, 1.23 ± 0.01 vs. 1.84 ± 0.01 g C kPa$^{0.5}$ kg$^{-1}$ H₂O, 0.13 ± 0.01 vs. −0.09 ± 0.006) (Fig. 4c). Overall, plants in the northern peatlands tended to adopt an "open" water-use strategy with lower uWUE and higher $PCOR_{Et \text{ vs. VPD}}$ in response to increasing VPD, leading to a weaker suppressive impact on vegetation growth compared to the global nonpeatland regions.

We further assessed the robustness of the satellite-derived differences in the plant traits between the northern peatlands and the global nonpeatland regions using eddy covariance flux towers. As the proxy for the plant water-use strategy, the uWUE in the northern peatlands was 0.78 g C kPa$^{0.5}$ kg$^{-1}$ H₂O lower than that in the global nonpeatland regions (mean ± 1 se, 1.10 ± 0.12 vs. 1.88 ± 0.10 g C kPa$^{0.5}$ kg$^{-1}$ H₂O) (Fig. 4c). Compared to the global nonpeatland regions, weak

stomatal regulation and an abundant atmospheric water supply in response to increasing VPD in the northern peatlands were also confirmed by the eddy covariance flux datasets. For stomatal activity, a significant negative response of $Gc$ to VPD was found in only 5.6% of the eddy covariance flux towers in the northern peatlands (coefficient mean ± 1 se, 0.04 ± 0.08), whereas this percentage increased to 52.6% in the global nonpeatland regions (− 0.27 ± 0.04) ($p < 0.05$, Fig. 4c, Methods). As a proxy for atmospheric water supply with increasing VPD, a significant negative response of evapotranspiration (ET) to VPD was observed in 36.8% of the towers in the global nonpeatland regions (coefficient mean ± 1 se, −0.16 ± 0.04), but this percentage was 11.1% in the northern peatlands (0.27 ± 0.08) (Supplementary Fig. 8).

In addition to the plant traits, the impact of VPD was also sensitive to the soil hydraulic properties and water availability. $PCOR_{GPP \text{ vs. VPD}}$

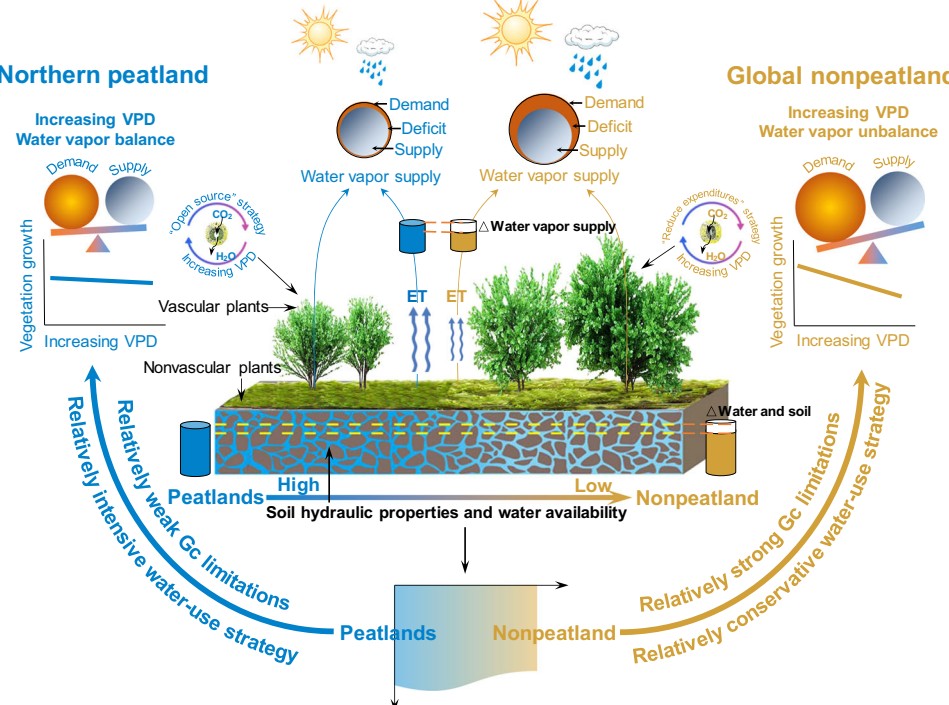

**Fig. 5 | Schematic illustrating divergent vegetation responses to rising vapor pressure deficit (VPD) between the northern peatlands and the global non-peatland regions.** Compared to the global nonpeatland regions, the supply of atmospheric water vapor, deriving from ample soil water availability and high coverage of nonvascular plants (e.g., moss), could meet the water demand of increasing VPD in the northern peatlands, as evidenced by slight changes in relative humidity (RH). In a water-rich environment of the northern peatlands, plants tend to adopt an "open" water-use strategy with increasing VPD, leading to a weak regulation of stomatal activity and, thus, neutral VPD impacts on vegetation growth. Both lines and approximate trapezoids with color gradients from blue to brown indicate water conditions and soil hydraulic properties from high (or good) to low (or weak). Both pairs of cylinders indicate the differences (white in the cylinders) in water vapor supply (top, △water vapor supply) and water conditions and soil hydraulic properties (bottom, △water and soil) between northern peat-lands and global nonpeatland regions. Differences in changes in RH between the northern peatlands and the global nonpeatland regions are exhibited by the balance in "supply" (orange circle) and "demand" (blue circle). The left (blue) and right (brown) of the figure indicate the northern peatlands and global nonpeatland regions, respectively.

increased with increasing VWC and SOC, but it decreased with increasing BD and CWD (Fig. 4b). Compared to the global nonpeatland regions, higher VWC (mean ± 1 se, 17.58 ± 0.12 vs. 13.56 ± 0.02%) and SOC (1.11 ± 0.02 vs. 0.26 ± 0.002%) and lower BD (1.06 ± 0.01 vs. 1.30 ± 0.001 g m$^{-3}$) and CWD (15.9 ± 0.20 vs. 37.4 ± 0.20 mm) were observed in the northern peatlands (Fig. 4c). These differences suggested that vegetation in the northern peatlands with good soil hydraulic properties and abundant water availability could resist the atmospheric water stress caused by increasing VPD compared to the global nonpeatland regions.

We then assessed the cascading correlations of plant water-use strategy, water availability, and soil hydraulic properties with VPD effects using mediation effect models (Methods). The results showed that the water availability and the soil hydraulic properties significantly influenced the VPD effects by determining the plant water-use strategy ($p < 0.05$, Fig. 4d, e). Specifically, changes in VWC (standardized indirect effect ± 1 se from three datasets of satellite-derived GPP, 0.14 ± 0.001) and SOC (0.15 ± 0.001) had a significant positive indirect effect on PCOR$_{GPP\ vs.\ VPD}$ through decreasing uWUE and increasing PCOR$_{Et\ vs.\ VPD}$ ($p < 0.05$, Fig. 4d, e). In contrast, a significant negative indirect effect was observed from changes in CWD ($-0.26 ± 0.01$) and BD ($-0.18 ± 0.001$) ($p < 0.05$, Fig. 4d, e). Collectively, compared to the global nonpeatland regions, plants in the northern peatlands adopted an "open" water-use strategy in response to increasing VPD because of the wet environment, favorable soil hydraulic properties, and adequate atmospheric water supply, resulting in a neutral response of vegetation growth to increasing VPD.

## Implications

This study represents one of the first attempts to mechanistically examine the VPD impacts on vegetation growth in northern peatlands. The neutral response of vegetation growth to warming-induced increasing VPD in the northern peatlands (Fig. 5) suggested that the prevailing views of vegetation suppression under increasing VPD are not necessarily the case across the globe. This can be explained by the fact that plants in the wet soil–air environment of the northern peatlands tended to adopt an "open" water-use strategy by relaxing stomatal regulation to maximize carbon uptake even as VPD increases (Fig. 5).

An ample atmospheric water supply, driven by a water-rich environment and high moss cover, was a critical factor in determining whether the increasing VPD was caused by warming alone in the northern peatlands. Factors contributing to the wet environment (e.g., relatively high VWC and low CWD) were the high water table[43], snow melt[44], permafrost thaw[45], and good soil hydraulic properties (e.g., relatively low BD and high SOC), which could directly accelerate water movement from the underlying surface into the atmosphere to meet the increasing water demand caused by rising VPD[28] in the northern peatlands. A high moss cover is another essential contributor to the atmospheric water supply with increasing VPD in the northern peatlands[18]. Mosses can store substantial water in their interconnected cavernous structures[46], yet they fail to minimize water loss under increasing atmospheric water demand due to a lack of stomatal regulatory structures[47,48]. Combining these characteristics with a large surface area, the water evaporation rate in moss is even higher than in

open water[47]. Under the same environmental conditions, our synthesized observations showed that the ET of moss was significantly greater by $0.43 \pm 0.14$ mm day$^{-1}$ (mean $\pm$ 1 se) than that of vascular plants ($p = 0.009$, Supplementary Fig. 9). The moss-dominated wet system of the northern peatlands allowed ET and Et to increase with increasing VPD[18]. Therefore, a sufficient atmospheric water supply allowed increases in AVP in approximately the same proportion as the SVP, leading to increasing VPD induced by warming alone in the northern peatlands.

A neutral response of vegetation growth to warming-induced increased VPD was observed in the northern peatlands. This is contrary to the prevailing views that increasing VPD induced by the coaction of warming and decreased RH markedly depressed vegetation growth in global nonpeatland regions[6–10,22,49]. A relatively dry environment in the global nonpeatland regions can limit atmospheric water supply under increasing VPD, disrupting the supply-demand balance of atmospheric water conditions and exacerbating atmospheric water stress[27]. This can increase the hydraulic burden of plants, limiting their stomatal activity to preventing excessive water loss at the expense of photosynthesis[16,50]. In contrast, the moss-dominated wet system of the northern peatlands can provide adequate atmospheric water to meet the increased water demand caused by increasing VPD. Even if atmospheric water stress occurs as warming-induced VPD increases, the stress may be below the threshold that leads to stomatal closure, as evidenced by the neutral response of Gc, Et, and GPP to increasing VPD. In this wet soil–air environment, plants adopt an "open" water-use strategy in response to water stress, maximizing carbon uptake by relaxing stomatal regulation[24,29]. Although an "open" water-use strategy may sacrifice hydraulic security, plants in water-rich environments would benefit more from keeping their stomata open to take up carbon than from conserving water[29,41]. Multisource datasets consistently demonstrated that plants in the northern peatlands were believed to resist increasing VPD driven by warming alone.

Three limitations should be acknowledged when interpreting the VPD effects. First, we used uWUE as a proxy for plant water-use strategy but not a more comprehensive metric to characterize vegetation response to water stress[29], such as stomatal safety margin (SSM), because of the data limitations at the grid scale. This alternative was supported by previous studies[40–42] and the positive correlation of uWUE with SSM[29] ($r = 0.61$, $p < 0.001$, Supplementary Fig. 10, Methods). Second, stomatal acclimation allows a high carbon assimilation rate in response to increased VPD, especially in species with an "open" water use strategy[19]. Although stomatal acclimation might be a possible mechanism for the neutral VPD effects in the northern peatlands[19], it has not been confirmed by our warming experiment in Mohe. Third, the comparison of the VPD effects was carried out between the northern peatlands and the global nonpeatland regions without a comprehensive analysis of spatial heterogeneity. A preliminary analysis found consistent VPD effects, and their underlying mechanisms (six plant traits and environmental factors, Supplementary Fig. 11) in the entire global nonpeatland regions were consistent with the nonhumid and humid regions. Future studies should be designed to address the spatial heterogeneity in VPD and its contributions to the VPD impacts on vegetation.

In conclusion, our study presented novel empirical evidence that VPD increases driven by warming alone did not necessarily suppress vegetation growth in the northern peatlands. The prevailing views of markedly decreasing vegetation growth with increasing VPD around the world might have overestimated the VPD suppression in the northern peatlands. The divergent VPD effects were caused by differences in water availability and soil hydraulic properties that regulated plant water-use strategy and stomatal activity in response to increasing VPD. Specifically, in global nonpeatland regions with relatively dry soils and air, plants adopted a "reduce expenditure" water-use strategy in response to increasing VPD and subsequently limited vegetation

growth. In the moss-dominated wet system of the northern peatlands, increasing VPD did not usually inhibit vegetation growth because plants evolved an "open" water-use strategy in the wet soil and air environment. Therefore, there is an urgent need to revisit and reframe how we represent vegetation growth under increased VPD in water-rich regions, such as northern peatlands, to accurately quantify and predict the land carbon sink.

## Methods

### Northern peatland maps
The northern peatlands are mainly distributed in the middle and high latitudes in the Northern Hemisphere ($> 30°$N), which contain more than 75% of the global peatlands[51,52]. The spatial distribution of the northern peatlands in this study was derived from PEATMAP[51]. The map is the result of a synthesized analysis of geospatial information from various sources at global, regional, and national scales[35] and has been used in a recent study[53]. The peatland extent in PEATMAP is extracted from Peat-ML, a map created using machine learning algorithms[52].

### Field warming experiments
First, we used a synthesized dataset to analyze the response of vegetation growth to increasing VPD from a recent meta-analysis[30]. Studies in the meta-analysis (1990–2020) were collected by searching Google Scholar and Web of Science. The keywords for the topic search were (a) "plant growth" OR "plant height" OR "plant abundance" OR "plant biomass" OR "belowground biomass" OR "aboveground biomass" OR "biomass" OR "production" OR "net primary productivity" OR "plant response" OR "biogeochemical process" and (b) "warming" OR "experimental warming" OR "elevated temperature" OR "climate change" and (c) "wetland" OR "wet tundra" OR "fen" OR "bog" OR "marsh" OR "swamp" OR "peatland"[30]. Selected publications followed these criteria: (1) experiments, including warming and control treatments, were conducted over at least one growing season; (2) peatlands could be distinguished as vascular plant-dominated (i.e., graminoids, shrubs) or cryptogam-dominated (i.e., mosses and lichens) based on the original site descriptions or relevant results (i.e., species abundance or biomass); and (3) studies included the variables of means, se, standard deviations and sample sizes in control and warming treatments[30]. The compiled database contained five response variables: plant abundance, plant height, aboveground net primary productivity, belowground net primary productivity, and NPP. A total of 273 independent sites from 51 studies were collected in the northern peatlands ($34.72°$N to $78.88°$N). In this study, we used the NPP to analyze the vegetation growth response to increasing VPD (67 sites, Supplementary Table 1).

VPD was missing from these 67 synthesized warming experiment sites. We extracted Ta and VPD from the CRU 4.04 datasets at the 67 warming experiment sites (reporting NPP) based on their geographic locations. Similar extractions of environmental variables from the CRU 4.04 datasets have also been reported in recent studies[6,31]. We found that the Ta and VPD from the CRU 4.04 datasets positively correlated with those from 18 eddy covariance flux towers, with a mean $r$ greater than 0.90. This good performance suggested that the CRU 4.04 datasets can be used to simulate the VPD in the 67 warming experiments. Furthermore, the eddy covariance flux towers and the CRU 4.04 datasets consistently showed that VPD exponentially increased with Ta, with a mean $r$ greater than 0.90 (Supplementary Table 1). Based on the exponential relationships between Ta and VPD as estimated by the CRU 4.04 datasets, we simulated the VPD in the control and warming treatments and then obtained increasing VPD with warming.

We then used the warming experiment sites where the VPD (or RH) had been observed under control and warming treatments to further analyze the response of vegetation growth to increasing VPD.

The warming experiment at the Mohe site was conducted in a peatland in the northern Greater Hinggan Mountains (Tuqiang Forestry Bureau in Mohe city, Heilongjiang Province; 52.93°N, 122.83°E). The peatland is characterized by a humid monsoon climate in a cold temperate zone with a mean annual temperature and precipitation of −3.9 °C and 450 mm, respectively. The growing season lasts for *c.* 120 days, from mid-May to mid-September. Four common native plant species in the plant community are *Sphagnum palustre* (SP), *Vaccinium uliginosum* (VU), *Ledum palustre* (LP), and *Carex globularis* (CG)[54].

Our field experiment in Mohe was set up in July 2020, and the warming treatment was placed in a transparent greenhouse (Supplementary Fig. 12). The control treatment was placed in an ambient environment near the transparent greenhouse. The control and warming treatments included 60 mesocosms (square plastic barrels) ($n = 30$ per treatment), and four monocultures (SP, CG, VU, and LP), three mixtures of two species (CG-VU, SP-VU, and SP-LP), two mixtures of three species (SP-VU-LP and CG-VU-LP), and one mixture of four species (SP-CG-VU-LP) were included in the control and warming treatments ($n = 3$ per species mixture). To simulate the natural plant community in our experimental site, one mixture of four species (SP-CG-VU-LP) was chosen in our study, and the control and warming treatments included 6 mesocosms (square plastic barrels) ($n = 3$ per treatment). The observed species and peat soil columns were transplanted from nearby peatlands into square plastic barrels (45 × 45 × 40 cm). To guarantee homogeneities in the plants and soils, we implemented the following control measurements[54]. (1) Plants with relatively uniform height and coverage were selected from nearby natural peatlands. On the same day, we cleaned the selected plants by removing soil particles under running water and then weighed these plants. We found that the wet weight of the selected plants showed no significant differences between the warming (mean ± 1 se, $476.2 \pm 21.2$ g m$^{-2}$) and control ($451.6 \pm 23.1$ g m$^{-2}$) treatments ($p = 0.477$, Supplementary Fig. 13). (2) The square plastic barrels were filled with peat soils (thickness of 30−35 cm) and manually collected in the peatlands from which the plants were sourced. Soil total carbon (TC, $346.0 \pm 9.1$ vs. $336.5 \pm 4.0$ mg g$^{-1}$, $p = 0.393$) and total nitrogen (TN, $27.8 \pm 0.5$ vs. $28.9 \pm 1.7$ mg g$^{-1}$, $p = 0.578$) showed insignificant differences between the warming and control treatments (Supplementary Fig. 13). Additionally, the water in the control treatment was entirely supplied by precipitation. For the warming treatment, precipitation was collected in a container and then evenly sprinkled on each square plastic barrel in the transparent greenhouse after precipitation events[54]. The mean soil moisture content (SMC, %) in the warming treatment decreased insignificantly by 2.0% ($33.3 \pm 0.5$ vs. $31.3 \pm 0.7$%, $p = 0.073$) compared to the control treatment in the growing season of 2021 (Supplementary Fig. 13).

At the Mohe site, TC and TN were detected using a TC-TN analyzer (Shimadzu, Tokyo, Japan). Ta, RH, VPD, and $G_c$ in the 2021 growing season were used to investigate the variability of the environmental variables and the response of vegetation growth (e.g., $G_c$) to increasing VPD. Ta and RH were measured by a portable temperature and humidity sensor (5TM, METER Group Inc.) attached to a data logger (EM50/G, METER Group Inc.). These indicators were recorded at 30 min intervals. We installed the two sensors in each treatment ($n = 2$ per treatment) and took their average. VPD was calculated from Ta and RH using Eq. (1).

$$VPD(hPa) = 6.11 \times \exp\left(\frac{17.27 \times Ta(°C)}{Ta(°C) + 273.3}\right) \times \left(1 - \frac{RH(\%)}{100}\right) \quad (1)$$

$G_c$, as a proxy for the vegetation growth of *Vaccinium uliginosum*, was measured by an AP4 Porometer (AP4, Delta-t, UK). We measured five leaves of *Vaccinium uliginosum* in each mesocosm (square plastic barrel) under the control and warming treatments. In the peak growing season, diurnal dynamic measurements (9 AM, 14 PM, 17 PM; China

Standard Time) were conducted on August 9 and 18, 2021. Ta and RH were simultaneously measured by an AP4 Porometer.

We further collected another 6 field warming experiments from the northern peatlands to test the robustness of the findings at the Mohe site (Supplementary Table 2). Studies for these field warming experiments were selected from a recent meta-analysis in the northern peatlands, as previously mentioned[30]. Studies in this analysis were only selected if Ta and RH (or VPD) and vegetation growth (i.e., biomass), in control and warming treatments, were provided. Following these criteria, 6 studies were included in our synthesized analyses.

### Satellite and eddy covariance flux datasets

To further investigate whether observations from the field experiments were appropriate for a regional scale, satellite data was used in this study. Monthly solar radiation was derived from the reanalysis products of ERA5-Land ($0.1° \times 0.1°$)[55]. The monthly wind speed was obtained from TerraClimate (1/24th degree)[36]. Monthly Ta, precipitation, and AVP were obtained from CRU 4.04 datasets (land; $0.5° \times 0.5°$)[56]. Monthly SVP and VPD ($0.5° \times 0.5°$) were calculated by Eqs. (2) and (3), respectively[6]. Recently, the CRU 4.04 datasets have been widely applied to analyze the temporal changes in the trends of VPD and its impact on vegetation growth[6,13,15,16,20]. Monthly RH was the ratio between AVP and SVP. The ratio of precipitation to potential evapotranspiration was defined as AI (arid and semiarid and dry sub-humid regions, AI < 0.65; humid, AI ≥ 0.65) (Global-AI_PET_v3)[57].

$$SVP(hPa) = 6.11 \times \exp\left(\frac{17.27 \times Ta(°C)}{273.3 + Ta(°C)}\right) \quad (2)$$

$$VPD(hPa) = SVP(hPa) - AVP(hPa) \quad (3)$$

Three satellite-derived GPP products were used to detect vegetation responses to VPD. The monthly Vegetation Photosynthesis Model (VPM) GPP dataset from 2001 to 2018 is based on an improved theory of light use efficiency and applies a state-of-the-art vegetation index gap-filling and smoothing algorithm and a separate treatment of C3/C4 photosynthetic pathways ($0.05° \times 0.05°$)[58]. It is driven by satellite data from the Moderate-resolution Imaging Spectroradiometer (MODIS) and climate data from the National Centers for Environmental Prediction (Reanalysis II). Solar-induced chlorophyll fluorescence observed by the Orbiting Carbon Observatory-2 (OCO-2) has offered unprecedented opportunities for monitoring land photosynthesis. GOSIF GPP from 2001 to 2018 is produced by GOSIF and linear relationships between solar-induced chlorophyll fluorescence and GPP to map GPP globally at a 0.05° spatial resolution and an 8-day time step[59]. The monthly FLUXCOM GPP product from 1982 to 2015 is produced by using machine learning algorithms to merge carbon flux measurements from FLUXNET eddy covariance flux towers with remote sensing and meteorological data ($0.5° \times 0.5°$)[60]. Three satellite-derived GPP products have been widely used to analyze VPD effects globally[13,20,21].

ET and Et were used to estimate the variables in the random forest models. The MOD16A2 Version 6 ET product is an 8-day composite dataset produced at 500 m pixel resolution. The algorithm of the MOD16 data product collection is based on the Penman–Monteith equation. Inputs included daily meteorological reanalysis data along with MODIS remotely sensed data products such as vegetation property dynamics, albedo, and land cover[61]. The dataset of Et was estimated by a coupled diagnostic biophysical model (Penman–Monteith-Leuning model, PML-v2, 500 m, 8 days)[62]. PML-v2 is developed by coupling the widely used photosynthesis model and a canopy stomatal conductance model with the Penman–Monteith energy balance equation[62]. The datasets mentioned above were aggregated to a spatial resolution of 0.5° and temporal resolution of 1 month from 1982 to 2018.

The flux tower-based GPP, latent heat flux (LE, W m$^{-2}$), sensible heat flux (H, W m$^{-2}$), and environmental variables of Ta, VPD, precipitation, shortwave radiation, wind speed (m s$^{-1}$), friction velocity (u$_8$, unitless), and atmospheric pressure were obtained from the global eddy-covariance flux dataset, FLUXNET2015 (global nonpeatland regions) and FLUXNET-CH$_4$ Community Product (northern peatlands). Following recent studies on VPD effects, we used GPP estimates based on the nighttime partitioning method (i.e., "GPP_NT_VUT_REF")[7,20]. We identified and used sites with at least 3 years (more than 15 months) of high-quality data (≥75% of good quality data in a month)[63]. In addition, we removed all cropland towers in the study area to exclude the effects of human activity[7]. Collectively, we compiled a database consisting of 113 flux tower sites (FLUXNET2015, 95 sites; FLUXNET-CH$_4$ Community Product, 18 sites). More detailed information on these sites is given in Supplementary Tables 3 and 4, including site name, site coordinates (latitude/longitude), start year of measurements and end year of measurements. The aforementioned variables from satellite and eddy-covariance flux datasets were chosen in the growing seasons from 1982 to 2018. Following a recent study[6], the growing season was defined as the month in which the mean Ta was greater than zero.

Flux tower-based $Gc$ (mm s$^{-1}$) was calculated by rearranging the Penman–Monteith equation using the following formula (Eqs. 4 and 5)[64,65].

$$Gc = \left[ \left( \frac{\Delta}{\gamma} \times \frac{H}{LE} - 1 \right) \times \gamma_a + \frac{\rho C_p}{\gamma} \times \frac{VPD}{LE} \right]^{-1} \quad (4)$$

where $\Delta$ is the ratio of the change in SVP to Ta (Pa K$^{-1}$); $\gamma$ is the psychometric constant (Pa K$^{-1}$); $\rho$ is the air density (kg m$^{-3}$); $C_p$ is the specific heat of air at constant pressure (J kg$^{-1}$ K$^{-1}$). Following a previous study[7] in Eq. 5, $\gamma_a$ is the aerodynamic resistance (s m$^{-1}$); $k$ is the von Kármán constant ($k = 0.4$); and $ws$ is the wind speed.

$$\gamma_a = (ws \times k^2) / \log \left( \exp \left( \frac{k \times ws}{u_8} \right) - 0.7 \right)^2 \quad (5)$$

uWUE was estimated by Eq. (6)[41]. ET (kg H$_2$O m$^{-2}$ s$^{-1}$) was converted from LE (W m$^{-2}$) with a coefficient of 2.44 J kg$^{-1}$ H$_2$O. To further demonstrate the uWUE as a proxy for plant water-use strategy, we collected the SSM (the difference between water potential at 88% loss of stomatal conductance and the water potential causing 50% loss of hydraulic conductivity) in 133 species in 44 sites from a recent meta-analysis[29] and found that uWUE significantly increased with increasing SSM ($r = 0.61$, $p < 0.001$, Supplementary Fig. 10). More detailed information on these SSM sites can be found in a recent study[29].

$$uWUE(gCkPa^{0.5}kg^{-1}H_2O) = \frac{GPP(gCm^{-2}) \times \sqrt{VPD(kPa)}}{ET(kgH_2Om^{-2})} \quad (6)$$

The flux tower-based data were used to evaluate the robustness of the VPD effects from the satellite-based datasets based on the PCOR analysis. According to the latitude, longitude, and time span from eddy covariance flux towers, we extracted grid environment variables, FLUXCOM GPP, VPM GPP, and GOSIF GPP. We then compared the VPD effects estimated from eddy covariance flux towers and satellite-based datasets. It should be noted that GOSIF GPP is completely independent of climate data; therefore, it might be more reliable than the other two products in assessing VPD impacts on vegetation growth.

## Variables in the random forest models
Six variables were used as predicative factors in the random forest models, including BD, SOC, CWD, VWC, uWUE, and PCOR$_{Et\ vs.\ VPD}$. The BD, SOC, and VWC are derived from the gridded Global Soil Dataset (GSD) based on the Soil Map of the World and various regional and national soil databases, including soil attribute data and soil maps[32].

The dataset is 30 arc seconds (-1 km at the equator). The vertical distributions of these soil properties ranged from 0 to 2.296 m (8 layers). We selected the soil properties in the 0-0.289 m (1 − 4 layers) and 0–1.383 m (1 − 7 layers) soil profiles for the grassland and forest (or shrubland), respectively. The land classifications were obtained from GLASS-GLC at 5 km resolution[66]. The available VWC differed between the VWC at −10 kPa (field capacity) and −1500 kPa (permanent wilting point) soil matric potential[33]. uWUE was estimated by CRU VPD and MODIS ET, GPP (Eq. 6). PCOR$_{Et\ vs.\ VPD}$ was assessed by PML Et and CRU VPD based on the PCOR analyses. CWD was derived from TerraClimate datasets and is calculated using a water balance model incorporating reference evapotranspiration, precipitation, temperature, and interpolated plant extractable soil water capacity[36].

## Synthesized comparative observations of ET between vascular plants and mosses
We compared ET between vascular plants and mosses and further investigated the mechanisms of the divergent drivers and impacts of increasing VPD. Studies included in this meta-analysis (1990−2022) were collected by searching Web of Science and Google Scholar for the following keywords: (a) "moss" OR "sphagnum" AND (b) "shrub" OR "graminoid" OR "vascular plant" AND (c) "evapotranspiration" OR "evaporation" OR "water loss". Selected publications had the following criteria: (1) ET was observed in the northern peatlands; (2) ET (mm day$^{-1}$) of vascular plants and mosses was simultaneously observed in the same environmental conditions; and (3) these observations were conducted for at least one growing season. Following these criteria, 16 comparative data pairs from 5 studies were included in this synthesized analysis (Supplementary Table 5).

## Statistical analyses
A paired $t$-test was applied to estimate the effects of warming on RH, Ta, and VPD at the Mohe site and the other 6 synthesized sites by the $t.test$ function and to assess the differences in ET between vascular plants and mosses. Shapiro–Wilk was calculated by the $shapiro.test$ function and was used to test the normal distributions of differences in these variables between the control and warming treatments. If study variables did not fit the normal distributions, we used a nonparametric test. Due to RH, Ta, and VPD being measured by two repetitions in each treatment at the Mohe site, we could not use the commonly accepted one-way analysis of variance to examine statistical differences in these variables between the control and warming treatments. Differences in biomass, TN, TC, and SMC between the control and warming treatments at the Mohe site were tested by one-way analysis of variance.

A least squares linear regression approach was used to detect temporal changes in the trends of Ta, RH, and VPD from 1982 to 2018 using the $lm$ function (Eq. 7).

$$y = \beta_0 + \beta_1 \times t + \varepsilon \quad (7)$$

where $y$ is Ta, RH, and VPD; $t$ is the year; $\beta_0$ and $\beta_1$ are the regression coefficients, and the investigated variable linear trend is $\beta_1$; and $\varepsilon$ is the residual of the fit. These coefficients were estimated by least squares linear regression.

We used PCOR analyses to assess the impacts of VPD on GPP, ET, and Et using satellite data and eddy-covariance flux data when Ta, precipitation, wind speed, and radiation were considered. In the PCOR analyses, all the variables were detrended by the $detrend$ function (trend type = "linear") in the $pracma$ package. We implemented the PCOR analyses using the $pcor.test$ function in the $ppcor$ package.

Random forest models were applied to parse the sensitivity of the VPD effects to plant traits and environmental factors using the $randomForest$ function in the $randomForest$ package. Sixty percent of the data were used to train the models, and the remaining 40% were used for validation. $r$ was used to assess the performance of the simulations,

and the mean $r$ was >0.8 in this study. In the analyses, one of the predictor variables was perturbed by one standard deviation (a value of 1 due to the initial input data normalization), and $PCOR_{GPP\ vs.\ VPD}$ was predicted again using the existing random forest model with the predictors including the perturbed ones; this process was repeated for each predictor variable. The predicted $PCOR_{GPP\ vs.\ VPD}$ was obtained with and without perturbation, and then we compared it to determine the differences in the sensitivity of the VPD effects to changes in the predictor variables. The calculation of VPD effect sensitivity to plant traits and environmental factors is shown in Eq. (8)[7,26]. Sensitivity analyses were repeated 15 times, and the median was used in our study.

$$Sensitivity_{VPD\,effects} =$$
$$median\left[\frac{Perturbed\ PCOR_{GPPvs.VPD} - No\ Perturbed\ PCOR_{GPPvs.VPD}}{stdev(No\ normalization\ PCOR_{GPPvs.VPD})}\right] \quad (8)$$

Mediation effect models based on the function *PROCESS* (*bruceR* package) were used to investigate the cascading correlations in which the water variability and soil hydraulic properties influenced the VPD effects by affecting the plant water-use strategy. The mediation effect model can test a hypothetical causal chain where one variable $X$ (independent variable) affects a second variable $M$ (mediator) and, in turn, that variable affects a third variable $Y$ (dependent variable). In this model, the independent variables were grid VWC and CWD (water availability), BD and SOC (soil hydraulic properties), the mediators were grid $PCOR_{Et\ vs.\ VPD}$, and uWUE (plant water-use strategy), and the dependent variable was grid $PCOR_{GPP\ vs.\ VPD}$ (VPD effects).

## Reporting summary
Further information on research design is available in the Nature Portfolio Reporting Summary linked to this article.

## Data availability
The datasets used in this study have been deposited in the public data repository (https://figshare.com/s/a1b39bfc3a2077cc0515) (Supplementary Table 6).

## Code availability
The data in this study were analyzed with publicly available tool packages in R 4.3.0. All the scripts for data analyses are available at https://figshare.com/s/a1b39bfc3a2077cc0515.

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

## Acknowledgements

The work was supported by the National Natural Science Foundation of China (Grant No. 42220104009 to C.S.; 42101109 to N.C.; 42271122 to L.S.), the Jilin Provincial Natural Science Foundation to N.C. (Grant No. YDZJ202201ZYTS477), and the China Postdoctoral Science Foundation-funded project to N.C. (Grant No. 2020M681058; 2022T150644). X.X. has been supported by the U.S. National Science Foundation (2145130).

## Author contributions

N.C. and X.X. designed the study. N.C. performed the analysis. Y.Z., C.S. and X.W. contributed the field data. N.C. and X.X. wrote the paper with input from all coauthors. F.Y., M.X., Q.W., G.H., T.B., Y.Z., J.L., T.Z., Y.S., L.S., Y.G., H.Z., G.M., and Y.D. provided method suggestions and contributed to the interpretation of the results.

## Competing interests

The authors declare no competing interests.
