## [Peer Review File · Nature Communications]

Reviewers' Comments:

Reviewer #1:

Remarks to the Author:

Chen et al. is another contribution to the ongoing debate on the relative role of soil moisture vs. vapor pressure deficit (VPD) in determining ecosystem-scale carbon/water fluxes. The authors make use of realistic warming experiments to show that rising VPD cannot depress peatland vegetation growth, which is quite convincing and the most exciting part of this study. In addition, the authors utilize partial correlation to estimate VPD impacts on vegetation at the grid-cell scale based on satellite monitoring of LAI/NDVI and model simulations. I have some concerns about this part. Do you detrend climate and vegetation data before calculating correlations? Could we trust models in representing VPD-Vegetation response? Probably not, in such a condition, how could we trust their projection in the future? For instance, in Figure 3d, along a warming gradient, VPD impacts on vegetation do not increase linearly but only become significantly negative in SSP585.

Overall, this is a useful contribution to this ongoing debate. I advise authors to focus on results from field experiments. I'm happy to review this MS again.

Other comments:

- Can you show comparisons in the VPD-vegetation response between field-scale and grid-scale (satellite or model)?
- Figure 2: Improve the ease of understanding. For instance, adding remarks of "Ta", etc into panels; what does the pie-chart of the panel (a) mean?
- I do not think this study has enough evidence to support Fig. 5.
- Define the "Variability".

Reviewer #2:

Remarks to the Author:

Chen et al. presented an interesting finding that warming-induced VPD increase will not significantly suppress peatland vegetation growth in the northern hemisphere. This challenges the prevailing opinion that future warming will increase VPD and thus impair vegetation growth. Chen et al.'s findings have the potential to impact the community. But before I recommend publication, I'd like to see how the authors address my comments.

1. The physiological process: I have difficulty understanding how plants sense the impact of VPD. Is there any physiological explanation for why vegetation can differentiate the impact between temperature and relative humidity?
2. The field measurement: The meat of this analysis is the field warming experiments. Unfortunately, the description of this experiment lacks sufficient details. First, I recommend that the authors insert a photo of the experiment set up at least as a supplementary figure. Second, how long was the warming applied? Will VPD acclimation play a factor here (<https://onlinelibrary.wiley.com/doi/10.1111/pce.12790>)? How do you control other meteorological conditions? How did the soil moisture level differ between control vs. treatment?
3. Line 126: Why did you remove Ta effect? In your case, your results seemed to indicate RH-induced VPD changes were not causing negative effects on NDVI and LAI? Moreover, why did you remove the radiation and precipitation effects here? There must be co-linearity between radiation, precipitation, and temperature. This did not mean temperature would not have a casual effect on vegetation growth.
4. Line 173: How did you predict that cryptogam loss and vascular plant expansion will increase in the future? I had a hard time understanding the purpose of your ESM analyses. ESMs are based on assumptions and most likely have the prevailing opinion of VPD negative impacts on vegetation hardwired in their algorithms. I did not think your results derived from ESMs will "support" your

observational analyses.

Other comments:

Line 105: the y axis of Fig. 1d and 1h were not canopy conductance.

Line 248: Unpublished data should be correctly referenced or added to the supplementary material.

Line 288: Why modeling center not field scientists?

Eq 1-6: Unit should be added.

Reviewer #1 (Remarks to the Author):

Chen et al. is another contribution to the ongoing debate on the relative role of soil moisture vs. vapor pressure deficit (VPD) in determining ecosystem-scale carbon/water fluxes. The authors make use of realistic warming experiments to show that rising VPD cannot depress peatland vegetation growth, which is quite convincing and the most exciting part of this study. Overall, this is a useful contribution to this ongoing debate. I advise authors to focus on results from field experiments. I'm happy to review this MS again.

Reply: We appreciate the reviewer for the very positive evaluation of the manuscript. In the revised manuscript, we further added 67 warming experiment sites in the northern peatlands. These synthesized warming experiments showed a neutral response of net primary productivity to increasing vapor pressure deficit (VPD) (coefficients ± 1 standard error, -0.02 ± 0.08 , $p = 0.769$). In addition, we used the eddy-covariance flux dataset, including FLUXNET2015 (global non-peatland regions, 95 sites) and FLUXNET-CH₄ Community Product (northern peatlands, 18 sites), to assess the robustness of our satellite-derived VPD effects. Based on the 18 eddy-covariance flux towers in the northern peatlands, we found that only 1 out of 18 towers showed a significant negative effect of VPD on gross primary productivity (GPP). As suggested by the reviewer, we have estimated and described the site-scale VPD effects in Results (synthesized warming experiments, Line 136–151; eddy-covariance flux, Line 239–247) and Methods (synthesized warming experiments, Line 438–467; eddy-covariance flux, Line 569–583).

In addition, the authors utilize partial correlation to estimate VPD impacts on vegetation at the grid-cell scale based on satellite monitoring of LAI/NDVI and model simulations. I have some concerns about this part. Do you detrend climate and vegetation data before calculating correlations?

Reply: Thank you for the suggestions. In the revised manuscript, we have de-trended GPP and the environmental variables of VPD, temperature (Ta), precipitation, wind speed, and radiation in all the partial correlation analyses (PCOR). The de-trended analyses were estimated by the *detrend* function (trend type = “linear”)¹ in *pracma* package in R 4.3.0. In our revised manuscript, we used satellite-derived GPP (VPM GPP, GOSIF GPP, and FLUXCOM GPP) to replace satellite-derived NDVI/LAI to quantify the VPD impacts on vegetation growth as GPP directly reflects the plants' ability to assimilate carbon dioxide into biomass. Following the reviewer's suggestions, we have described the de-trended analyses in the Methods (see lines 658–662 for the detailed description).

Could we trust models in representing VPD-Vegetation response? Probably not, in such a condition, how could we trust their projection in the future? For instance, in Figure 3d, along a warming gradient, VPD impacts on vegetation do not increase linearly but only become significantly negative in SSP585.

Reply: Thank you for raising this important point. In line with reviewers' concerns, recent studies have consistently shown that ESMs underestimate the negative effect of VPD on vegetation growth compared to the eddy-covariance flux towers². These uncertainties may be due to the lack of consideration of VPD effect algorithms³ and mismatching vegetation classification in the northern peatlands^{4, 5} in current ESMs. Therefore, we removed the ESMs in our revised manuscript and mainly focused on the VPD effects estimated from multi-source datasets of in situ observations, multi-site synthesis (78 sites), eddy-covariance flux towers (113 sites; FLUXNET-

CH4 Community Product, 18 sites; FLUXNET2015, 95 sites), and regional scale remoting sensing products.

The initial objectives for using the ESMs were to investigate whether the response of vegetation growth to increasing VPD varied with the drivers of increasing VPD (warming alone vs. the covariation of warming and decreasing relative humidity, RH), and to determine whether the current prevailing view of a marked decline in vegetation growth with increasing VPD can represent the VPD effects of the northern peatlands. As an alternative to the ESMs in our revised manuscript, we compared the VPD effects between the northern peatlands and the global non-peatland regions based on the eddy-covariance flux dataset and satellite-derived dataset. These comparisons could be more aligned with our original objective due to the prevailing views mainly derived from the global non-peatland regions^{6, 7, 8}. Our multi-source data implied that the suppression impact of increased VPD, driven by warming and decreasing RH co-varied, on vegetation growth in the global non-peatland regions was more potent than that caused by warming alone in the northern peatlands (Fig.1). This suggested that the relatively well-studied, markedly negative response of vegetation growth to increasing VPD induced by the coaction of warming and decreasing RH^{3, 9} could overestimate the suppression impact of warming-induced increasing VPD, at least in the northern peatlands. In our revised manuscript, we have added the comparisons in the drivers and effects of increasing VPD between the global non-peatland regions and the northern peatlands in Results (Line 201–256).

Fig.1 Changing rates of air temperature (Ta), relative humidity (RH), and vapor pressure deficit (VPD), and the impact of VPD on vegetation growth in the global non-peatland

regions. a–c: Changing rates of Ta, RH, and VPD from 1982 to 2018 in the global non-peatland regions. **d:** Differences in VPD effects as estimated by the mean $PCOR_{GPP\ vs.\ VPD}$ (horizontal solid arrow direction) and the percentage (%) of significantly negative $PCOR_{GPP\ vs.\ VPD}$ (horizontal dotted arrow direction) between the northern peatlands (blue) and the global non-peatland regions (brown). “N” indicates the number of eddy-covariance flux towers. Solid lines in the violin plots indicate the mean $PCOR_{GPP\ vs.\ VPD}$.

Other comments:

- Can you show comparisons in the VPD-vegetation response between field-scale and grid-scale (satellite or model)?

Reply: Thank you for the suggestions. Following the reviewer’s suggestions, we have compared the VPD effects between the eddy-covariance flux towers and the grid-scale of satellite datasets in Results (Lines 237–256) and Methods (Lines 606–610). PCOR analyses showed that the de-trended VPD was significantly negatively correlated with de-trended GPP at 1 out of 18 (5.6%) and 43 out of 95 (45.3%) eddy-covariance flux towers in the northern peatlands and the global non-peatland regions, respectively ($p < 0.05$, Fig.1d). Grouping the 95 eddy-covariance flux towers into the non-humid regions (33 towers) and the humid regions (62 towers) further confirmed a greater VPD suppression impact in the global non-peatland regions than the northern peatlands, with a percentage of significantly negative $PCOR_{GPP\ vs.\ VPD}$ of 63.6% and 35.5% in the non-humid regions and the humid regions, respectively (Supplementary Fig.6). In addition, according to the latitude, longitude, and time span in 113 eddy-covariance flux towers, we found that the symbols of the satellite-derived $PCOR_{GPP\ vs.\ VPD}$ were in agreement with 76.4% (mean value from three satellite-derived GPP) of the eddy-covariance flux towers (Fig.2). The satellite-derived $PCOR_{GPP\ vs.\ VPD}$ positively correlated with eddy-covariance $PCOR_{GPP\ vs.\ VPD}$, with a correlation coefficient of 0.51 to 0.58 ($p < 0.05$, Fig.2). Collectedly, the field- and grid-scale observations consistently suggested that the current prevailing views from the global non-peatland regions may overestimate the VPD suppression impact in the northern peatlands.

Fig. 2 Correlations of eddy-covariance $PCOR_{GPP\ vs.\ VPD}$ with satellite-derived $PCOR_{GPP\ vs.\ VPD}$ that estimated by FLUXCOM GPP, VPM GPP, and GOSIF GPP. “r” indicates the correlation coefficient. $PCOR_{GPP\ vs.\ VPD}$ in the 1st and 3rd quadrants show the same symbols between eddy

covariance $PCOR_{GPP \text{ vs. } VPD}$ and satellite-derived $PCOR_{GPP \text{ vs. } VPD}$, 2nd and 4th quadrants show opposite symbols.

- **Figure 2: Improve the ease of understanding. For instance, adding remarks of “Ta”, etc into panels; what does the pie-chart of the panel (a) mean?**

Reply: Thank you for the suggestions. We used the histogram to replace the pie chart, showing the percentage of the response of GPP to VPD (Fig.3). The names of satellite-derived GPP used to assess the VPD effects were marked at the top of the insets (Fig.3). More text has been added as notes to the panels (Fig.3).

Fig.3 Changing rates of air temperature (T_a), relative humidity (RH), vapor pressure deficit (VPD); spatial distributions of the VPD impacts on vegetation growth across the northern peatlands during 1982–2018. a–c: Changing rates of T_a , RH, and VPD from 1982 to 2018. Solid and dotted lines indicate the significant and insignificant variability of T_a , RH, and VPD, respectively. Shading areas along the regression lines are the 95% confidence intervals. **d–f:** The spatial distributions of the $PCOR_{GPP \text{ vs. } VPD}$ derived from FLUXCOM GPP, GOSIF GPP, and VPM GPP. Insets show the percentage (%) of significant (Neg*) and insignificant (Neg) negative (brown) $PCOR_{GPP \text{ vs. } VPD}$, and significant (Pos*) and insignificant (Pos) positive (blue) $PCOR_{GPP \text{ vs. } VPD}$. **g–i:** Changes in regional mean $PCOR_{GPP \text{ vs. } VPD}$ (g–i) and the percentage of significant negative $PCOR_{GPP \text{ vs. } VPD}$ (j–l) across the peatland extent.

- I do not think this study has enough evidence to support Fig. 5.

Reply: Thanks for pointing this out. Following the reviewer's suggestions, we improved and added some processes in the updated schematic in the revised manuscript. The schematic was used to illustrate the drivers and effects of increasing VPD and determine their underlying mechanisms in the northern peatlands and the global non-peatland regions.

We identified the key plant traits of plant water-use strategy (underlying water use efficiency, $uWUE$; the response of vegetation transpiration to VPD, $PCOR_{Et \text{ vs. VPD}}$), the environmental factors of water availability (climate water deficit, CWD; available volumetric water content, VWC), and soil hydraulic properties (soil organic carbon, SOC; bulk density, BD) to explain the divergent VPD effects using random forest models (Fig.4). In addition, the water vapor supply and canopy conductance (G_c) were estimated by the response of evapotranspiration to VPD and the rearranging Penman-Monteith equation, respectively, based on the eddy-covariance flux datasets (Fig.4). Compared to the global non-peatland regions, higher $PCOR_{Et \text{ vs. VPD}}$, VWC, and SOC, and lower $uWUE$, CWD, and BD were observed in the northern peatlands. This suggested that the northern peatlands had adequate water availability, good soil hydraulic properties, and an "open" water-use strategy compared to the global non-peatland regions (Fig.4c). These differences were exhibited by both the line and approximate trapezoid with color gradients from the blue to the brown indicating water conditions and soil hydraulic properties from high (or good) to low (or weak) in Fig.5. Both pairs of cylinders showed the differences (white in the cylinders) in water vapor supply (top, Δ water vapor supply), and water availability and soil hydraulic properties (bottom, Δ water, and soil) between the northern peatlands and the global non-peatland regions (Fig.5). Detailed descriptions of the differences in these variables estimated by the site- and grid-scale datasets between the northern peatland and the global non-peatland regions are shown in the Result and Discussion section (Lines 258–322) and the Method section (Lines 612–625).

Combining with the sensitivity of the VPD effects to these variables using random forest models, we further analyzed the cascading correlations among these plant traits and environmental factors using the mediation effect models (Fig.4b,d,e; Results, Line 324–337; Methods, Line 664–686). Compared to the global non-peatland regions, plants evolved an "open" water-use strategy in response to increasing VPD in the northern peatlands because of the wet environment, good soil hydraulic properties, and adequate atmospheric water supply, resulting in a neutral response of vegetation growth to increasing VPD (Fig.4d,e). The differences in the VPD effects were shown by the correlations between "Increased VPD" and "Vegetation growth" in Fig.5. In addition, adequate atmospheric water supply in the northern peatlands contributed by the wet environment, good soil hydraulic properties, and high moss cover maintained the balance between supply and demand for atmospheric water (unchanged RH) with increasing VPD compared to a significant decrease in RH in the global non-peatland regions. Differences in RH between the northern peatlands and the global non-peatland regions were exhibited by the balance in "supply" (orange circle) and "demand" (blue circle) in the Fig.5.

Fig.4 Mechanisms for the divergent VPD effects between the northern peatlands and the global non-peatland regions. **a:** Correlations of observed $PCOR_{GPP \text{ vs. } VPD}$ from three satellite-derived GPP datasets with simulated $PCOR_{GPP \text{ vs. } VPD}$. **b:** Sensitives (mean ± 1 se; positive: blue histograms; negative: brown histograms) of the $PCOR_{GPP \text{ vs. } VPD}$ to water availability (VWC and CWD; blue dots), soil hydraulic properties (SOC and BD; grey dots), and plant water-use strategy ($PCOR_{Et \text{ vs. } VPD}$ and uWUE; green dots). **c:** Differences in water availability, soil hydraulic properties, and plant water-use strategy between the northern peatlands and the global non-peatland regions, including satellite-derived observations (left of dotted line) and eddy-covariance flux towers (right of dotted line). Solid lines in the violin plots indicate the mean of these variables. **d:** Schematic diagram illustrates the mediation effect models. Ind_X_M1_Y, Ind_X_M2_Y, and Ind_X_M1_M2_Y indicate that water availability (X), and soil hydraulic properties (X) influence the VPD effects (Y, GPP vs. VPD) through uWUE (M₁, Ind_X_M1_Y), $PCOR_{Et \text{ vs. } VPD}$ (Et vs. VPD; M₂, Ind_X_M2_Y), and both (Ind_X_M1_M2_Y). **e:** Standardized direct and indirect effects (mean ± 1 se) of water availability, and soil hydraulic properties on the VPD effects. “*” denotes that all the standardized effects are significant ($p < 0.05$).

Fig.5 Schematic illustrating divergent vegetation responses to rising vapor pressure deficit (VPD) between the northern peatlands and the global non-peatland regions. Compared to the global non-peatland regions, the supply of atmospheric water vapor, deriving from ample soil water availability and high coverage of no-vascular plants (e.g., moss), could meet the water demand of increasing VPD in the northern peatlands, as evidenced by slight changes in relative humidity (RH). In a water-rich environment of the northern peatlands, plants tend to adopt an “open” water-use strategy with increasing VPD, leading to a weak limitation of stomatal activity and, thus, the neutral VPD impacts on vegetation growth. Both line and approximate trapezoid with color gradients from blue to brown indicate water conditions and soil hydraulic properties from high (or good) to low (or weak). Both pairs of cylinders indicate the differences (white in the cylinders) in water vapor supply (top, Δ water vapor supply), and water conditions and soil hydraulic properties (bottom, Δ water, and soil) between northern peatlands and global non-peatland regions. Differences in changes in RH between the northern peatlands and the global non-peatland regions are exhibited by the balance in “supply” (orange circle) and “demand” (blue circle). The left (blue) and right (brown) of the figure indicate the northern peatlands and global non-peatland regions.

- Define the “Variability”.

Reply: Thanks for pointing this out. The variability was the temporal change in T_a , RH, and VPD trends from 1982 to 2018. A least squares linear regression approach was used to detect temporal changes in the trends of T_a , RH, and VPD³ using *lm* function in R 4.3.0 (Equation 1, Lines 649–656).

$$y = \beta_0 + \beta_1 \times t + \varepsilon \quad (1)$$

Where y was T_a , RH, and VPD; t was the year; β_0 and β_1 were the regression coefficients, and the investigated variable linear trend was β_1 ; ε was the residual of the fit. These coefficients were estimated by least squares linear regression.

Reviewer #2 (Remarks to the Author):

Chen et al. presented an interesting finding that warming-induced VPD increase will not significantly suppress peatland vegetation growth in the northern hemisphere. This challenges the prevailing opinion that future warming will increase VPD and thus impair vegetation growth. Chen et al.'s findings have the potential to impact the community. But before I recommend publication, I'd like to see how the authors address my comments.

Thank you for reviewing our paper and for your constructive suggestions.

1. The physiological process: I have difficulty understanding how plants sense the impact of VPD. Is there any physiological explanation for why vegetation can differentiate the impact between temperature and relative humidity?

Reply: Thanks for pointing this out. Increasing atmospheric demand for water induced by increasing VPD affects photosynthesis and transpiration of plants through controlling stomata activity and xylem conductance, and therefore plays a critical role in regulating water and carbon cycles of terrestrial ecosystems^{7, 8, 10}. As an indicative of atmospheric aridity, with increasing VPD, plants tend to close their stomata to minimize water loss and avoid critical water tension within the xylem at the cost of reducing or ceasing photosynthesis^{2, 8, 10, 11}. This negative response usually occurs with increasing VPD caused by increasing T_a and declining RH, covaried^{3, 9}. This may be because a dry water environment in the global non-peatland regions relative to the northern peatlands limited atmospheric water supply by increasing VPD, disrupting the supply-demand balance of atmospheric water conditions and exacerbating atmospheric water stress¹². The relatively dry soil and air environment could increase the hydraulic burden of plants, causing them to adopt a water conservative strategy in response to increasing VPD, limiting their stomata activity to prevent excessive water loss at the expense of reduced photosynthesis and suppressed vegetation growth^{10, 13}. These findings prevail in the current studies across the global non-peatland regions^{1, 2, 3, 7, 8, 14, 15, 16, 17}.

In contrast, an ample atmospheric water supply, driven by a water-rich environment and high moss cover, was a critical factor in determining the neutral response of vegetation growth to increasing VPD caused by warming alone in the northern peatlands. The wet environment (e.g., relatively high VWC and low CWD) in the northern peatlands contributed by the shallow water table¹⁸, snow melts¹⁹, and permafrost thaw²⁰, and good soil hydraulic properties (e.g., relatively low BD and high SOC). A high moss cover is another essential contributor to the atmospheric water supply with increasing VPD in the northern peatlands⁴. The moss-dominated wet system of the northern peatlands can provide adequate atmospheric water supply to meet the increased water demand caused by increasing VPD, as evidenced by the slight changes in RH, suggesting that increasing VPD is induced by warming alone. Even if atmospheric water stress occurs as increasing VPD induced by warming alone, the stress may be below the threshold that leads to stoma closure, as evidenced by the neutral response of G_c , E_t and GPP to increasing VPD. In the wet soil-air environment, plants adopt an “open” water-use strategy in response to water stress, maximizing carbon uptake by loosening stomatal regulation^{21, 22}. Our mediation effect models further supported that, compared to the global non-peatland regions, plants in the northern peatlands evolved an “open” water-use strategy in response to increasing VPD due to the wet environment, good soil hydraulic properties, and adequate atmospheric water supply, resulting in a marginal VPD suppression (Results, Lines

258–337). Multi-source datasets consistently demonstrated that plants in the northern peatlands were believed to resist increasing VPD driven by warming alone.

Despite that, three limitations should be acknowledged when interpreting the VPD effects (see lines 396–412 for the detailed description). Firstly, we used uWUE as a proxy for plant water-use strategy but not a more comprehensive metric to characterize vegetation response to water stress²¹, such as stomatal safety margin, because of the data limitations at the grid scale. Second, although the stomatal acclimation might be a possible mechanism for the neutral VPD effects in the northern peatlands²³, it has not been confirmed by our warming experiment in Mohe. Third, the comparison of the VPD effects was carried out between the northern peatlands and the global non-peatland regions without deep analysis of spatial heterogeneity. Future studies should be designed to tackle the spatial heterogeneity in VPD and its VPD effects.

2. The field measurement: The meat of this analysis is the field warming experiments. Unfortunately, the description of this experiment lacks sufficient details. First, I recommend that the authors insert a photo of the experiment set up at least as a supplementary figure. Second, how long was the warming applied? Will VPD acclimation play a factor here (<https://onlinelibrary.wiley.com/doi/10.1111/pce.12790>)? How do you control other meteorological conditions? How did the soil moisture level differ between control vs. treatment?

Reply: We thank the reviewer for this important point. Following the reviewer's suggestions, we have improved the description of this experiment in the revised manuscript. We added a photo of the warming experiment at the Mohe site (Fig.6). Our field experiment in Mohe was set up in July 2020, and the warming treatment was placed in a transparent greenhouse. The control treatment was placed in a natural environment near the transparent greenhouse. To simulate the natural plant community, the control and warming treatments included 6 mesocosms (square plastic barrels) ($N = 3$ per treatment), and the common native plant species of *Sphagnum palustre*, *Vaccinium uliginosum*, *Ledum palustre*, and *Carex globularis* were transplanted into each square plastic barrel. Observed species and peat soil columns were transplanted from nearby peatlands into square plastic barrels ($45 \times 45 \times 40$ cm). To guarantee homogeneities in the plants and soils, we implemented the following control measurements²⁴. (1) plants with relatively uniform height and coverage were selected from nearby natural peatlands. On the same day, we cleaned the selected plants by removing soil particles under running water and then weighed these plants. We found that the wet weight of the selected plants showed no significant differences between warming (mean ± 1 standard error, $476.2 \pm 21.2 \text{ g m}^{-2}$) and control ($451.6 \pm 23.1 \text{ g m}^{-2}$) treatments ($p = 0.477$, Fig.7). (2) the square plastic barrels were filled with peat soils (thickness of 30–35cm), manually collected in the peatlands from which the plants were sourced. Soil total carbon (TC, 346.0 ± 9.1 vs. $336.5 \pm 4.0 \text{ mg g}^{-1}$, $p = 0.393$) and total nitrogen (TN, 27.8 ± 0.5 vs. $28.9 \pm 1.7 \text{ mg g}^{-1}$, $p = 0.578$) showed insignificant differences between warming and control treatments (Fig.7). Additionally, the water in the control treatment was entirely supplied by natural precipitation. For the warming treatment, precipitation was collected in a container and then evenly sprinkled on each square plastic barrel in the transparent greenhouse after precipitation events²⁴. Mean soil moisture content (SMC, %) in the warming treatment decreased insignificantly by 2.0% (33.3 ± 0.5 vs. $31.3 \pm 0.7\%$, $p = 0.073$)

compared to the control treatment in the growing season of 2021 (Fig.7). (See lines 469–520 for the detailed description).

Thanks for the reviewer providing a new view for explaining the VPD effects in the northern peatlands. Stomatal acclimation may also influence the VPD effects²³. Acclimation is commonly interpreted as an adaptive response that allows organisms to maintain homeostasis under changing environmental conditions^{25, 26}. Stomatal acclimation allows a high carbon assimilation rate in response to increased VPD but not the expected limited stomatal activity with increasing VPD²³. The acclimation response is more prevalent for species with evolving an “open” water-use strategy²³. The acclimation response could not be confirmed by our short-term warming experiment in Mohe (2020–2021). It is certain that the acclimation response can provide an additional explanation for the neutral suppression impact of warming-induced increased VPD on vegetation growth in the northern peatlands. This limitation warrants future studies using more long-term experiments to validate the acclimation response presented here.

Fig.6 The warming experiment at the Mohe site. SP, CG, VU, LP indicate *Sphagnum palustre*, *Carex globularis*, *Vaccinium uliginosum*, and *Ledum palustre*. The warming treatment is placed in a transparent greenhouse, and the control treatment is placed in a natural environment near the transparent greenhouse. Control and warming treatments included 60 mesocosms (square plastic barrels) (N=30 per treatment), including included four monocultures (SP, CG, VU, and LP), three mixtures of two species (CG-VU, SP-VU, and SP-LP), two mixtures of three species (SP-VU-LP and CG-VU-LP), and one mixture of four species (SP-CG-VU-LP) (N=3 per species mixture). To simulate the natural plant community in our experiment site, one mixture of four species (SP-CG-VU-LP) was chosen in our study, and the control and warming treatments included 6 mesocosms (square plastic barrels) (N = 3 per treatment).

Fig.7 Biomass, soil total carbon, soil total nitrogen and soil moisture content under the control and warming treatments at the Mohe site (error bars represent one unit of standard deviation). The same letter on top of the histograms indicates insignificant differences between control and warming treatments ($p > 0.05$).

3. Line 126: Why did you remove Ta effect? In your case, your results seemed to indicate RH-induced VPD changes were not causing negative effects on NDVI and LAI? Moreover, why did you remove the radiation and precipitation effects here? There must be co-linearity between radiation, precipitation, and temperature. This did not mean temperature would not have a casual effect on vegetation growth.

Reply: We thank the reviewer for this critical point. We fully agree reviewer's opinions. This is mainly attributable to our inaccurate representation of PCOR analyses. We have rewritten this part as below.

Three de-trended satellite-derived GPP positively correlated with de-trended VPD over 58.2 to 66.4% (24.8 to 33.0% with a significant positive correlation, $p < 0.05$) of the northern peatlands when de-trended Ta, radiation, wind speed, and precipitation were considered.

4. Line 173: How did you predict that cryptogam loss and vascular plant expansion will increase in the future? I had a hard time understanding the purpose of your ESM analyses. ESMs are based on assumptions and most likely have the prevailing opinion of VPD negative impacts on vegetation hardwired in their algorithms. I did not think your results derived from ESMs will “support” your observational analyses.

Reply: We thank the reviewer for this critical point. In our original manuscript, based on the synthesized observations from 273 independent sites, potential correlations of the effect size of warming on vascular plant aboveground net primary productivity (ANPP) and cryptogam ANPP with increases in Ta were evaluated using logarithmic functions²⁷. Based on the fitting modes, we assessed the effect size with current warming of 0.87 °C and projected warming of SSP126, SSP245, and SSP585 by the end of 2100. Then, we analyzed the temporal dynamic of the effect size of vascular plant ANPP and cryptogam ANPP.

In line with reviewers' concerns, recent studies have consistently shown that ESMs underestimate the negative effect of VPD on vegetation growth compared to the eddy-covariance flux towers². These uncertainties may be due to the lack of consideration of VPD effect algorithms³ and mismatching vegetation classification (i.e., the northern peatlands^{4, 5}) in current ESMs. Therefore, we have removed the analyses with ESM outputs in our revised manuscript and mainly focused on the VPD effects estimated from multi-source datasets of in situ observations, multi-site synthesis (78 sites), eddy-covariance flux towers (113 sites; FLUXNET-CH4 Community Product, 18 sites; FLUXNET2015, 95 sites), and regional scale remoting sensing products.

The initial objectives for using the ESMs were to investigate whether the response of vegetation growth to increasing VPD varied with the drivers of increasing VPD (warming alone vs. the covariation of warming and decreasing relative humidity, RH). Furthermore, to further determine whether the current prevailing view of a marked decline in vegetation growth with increasing VPD can represent the VPD effects of the northern peatlands. As an alternative to ESMs in our revised manuscript, we compared the VPD effects between the northern peatlands and the global non-peatland regions based on the eddy-covariance flux dataset and satellite-derived dataset. These comparisons could be more in line with our original objective due to the current prevailing views mainly derived from the global non-peatland regions^{6, 7, 8}.

Our multi-source data implied that the suppression impact of increased VPD, driven by warming and decreasing RH co-varied, on vegetation growth in the global non-peatland regions was more potent than that caused by warming alone in the northern peatlands (Fig.1). This suggested that the relatively well-studied, markedly negative response of vegetation growth to increasing VPD induced by the coaction of warming and decreasing RH^{3,9} could overestimate the suppression impact of warming-induced increasing VPD, at least in the northern peatlands. In our revised manuscript, we have added the comparisons in the drivers and effects of increasing VPD between the global non-peatland regions and the northern peatlands in Results (Line 201–256).

Other comments:

Line 105: the y-axis of Fig. 1d and 1h were not canopy conductance.

Reply: We thank the reviewer for this critical point. To be consistent with the analyses of synthesized warming experiments, we calculated the ratio of the canopy conductance (G_c) of *Vaccinium uliginosum* between control and warming treatments.

Line 248: Unpublished data should be correctly referenced or added to the supplementary material.

Reply: Thank you for your suggestions. We have added the spatial distribution of the vegetation transpiration in response to increasing VPD based on the PCOR analyses to the data availability link of <https://figshare.com/s/a1b39bfc3a2077cc0515>.

Line 288: Why modeling center not field scientists?

Reply: Thanks for your suggestions. This is mainly attributable to our inaccurate representation. In the revised manuscript, we have rewritten similar expressions as below. Future studies should be designed to tackle the spatial heterogeneity in VPD and its VPD effects.

Eq 1-6: Unit should be added.

Reply: Thanks for your suggestions. We have added the unit in the equations in the revised manuscript.

References

1. He B, *et al.* Worldwide impacts of atmospheric vapor pressure deficit on the interannual variability of terrestrial carbon sinks. *National Science Review* **9**, nwab150 (2022).
2. Fu Z, *et al.* Atmospheric dryness reduces photosynthesis along a large range of soil water deficits. *Nature Communications* **13**, 989 (2022).
3. Yuan W, *et al.* Increased atmospheric vapor pressure deficit reduces global vegetation growth. *Science Advances* **5**, eaax1396 (2019).
4. Helbig M, *et al.* Increasing contribution of peatlands to boreal evapotranspiration in a warming climate. *Nature Climate Change* **10**, 555-560 (2020).
5. Melton JR, *et al.* A map of global peatland extent created using machine learning (Peat-ML). *Geoscientific Model Development* **15**, 4709-4738 (2022).
6. Ding J, *et al.* Increasingly important role of atmospheric aridity on Tibetan alpine grasslands. *Geophysical Research Letters* **45**, 2852-2859 (2018).
7. Novick KA, *et al.* The increasing importance of atmospheric demand for ecosystem water and carbon fluxes. *Nature Climate Change* **6**, 1023-1027 (2016).
8. Konings AG, Williams AP, Gentine P. Sensitivity of grassland productivity to aridity controlled by stomatal and xylem regulation. *Nature Geoscience* **10**, 284-288 (2017).
9. Cheng Y, *et al.* A shift in the dominant role of atmospheric vapor pressure deficit and soil moisture on vegetation greening in China. *Journal of Hydrology* **615**, 128680 (2022).
10. Grossiord C, *et al.* Plant responses to rising vapor pressure deficit. *New Phytologist* **226**, 1550-1566 (2020).
11. Running SW. Environmental control of leaf water conductance in conifers. *Canadian Journal of Forest Research* **6**, 104-112 (1976).
12. Martin J, *et al.* Recent decline in the global land evapotranspiration trend due to limited moisture supply. *Nature* **467**, 951-954 (2010).
13. Oren R, *et al.* Survey and synthesis of intra- and interspecific variation in stomatal sensitivity to vapour pressure deficit. *Plant, Cell & Environment* **22**, 1515-1526 (1999).
14. Kimm H, *et al.* Redefining droughts for the U.S. Corn Belt: The dominant role of atmospheric vapor pressure deficit over soil moisture in regulating stomatal behavior of Maize and Soybean. *Agricultural and Forest Meteorology* **287**, 107930 (2020).
15. Liu L, Gudmundsson L, Hauser M, Qin D, Li S, Seneviratne SI. Soil moisture dominates dryness stress on ecosystem production globally. *Nature Communications* **11**, 4892 (2020).
16. Sulman BN, Roman DT, Yi K, Wang L, Phillips RP, Novick KA. High atmospheric demand for water can limit forest carbon uptake and transpiration as severely as dry soil. *Geophysical Research Letters* **43**, 9686-9695 (2016).
17. Chen N, *et al.* Multiple-scale negative impacts of warming on ecosystem carbon use efficiency across the Tibetan Plateau grasslands. *Global Ecology and Biogeography* **30**, 398-413 (2020).
18. Carpino OA, Berg AA, Quinton WL, Adams JR. Climate change and permafrost thaw-induced boreal forest loss in northwestern Canada. *Environmental Research Letters* **13**, 084018 (2018).
19. Qi W, Feng L, Liu J, Yang H. Snow as an important natural reservoir for runoff and soil moisture in Northeast China. *Journal of Geophysical Research: Atmospheres* **125**, e2020JD033086 (2020).
20. Helbig M, *et al.* Regional atmospheric cooling and wetting effect of permafrost thaw-induced boreal forest loss. *Global Change Biology* **22**, 4048-4066 (2016).
21. Jin Y, *et al.* Aridity-dependent sequence of water potentials for stomatal closure and hydraulic dysfunctions in woody plants. *Global Change Biology* **29**, 2030-2040 (2023).
22. Rogiers SY, *et al.* Stomatal response of an anisohydric grapevine cultivar to evaporative demand, available soil moisture and abscisic acid. *Tree Physiology* **32**, 249-261 (2012).
23. Marchin RM, Broadhead AA, Bostic LE, Dunn RR, Hoffmann WA. Stomatal acclimation to vapour pressure deficit doubles transpiration of small tree seedlings with warming. *Plant, Cell & Environment* **39**, 2221-2234 (2016).
24. Zhang Y, *et al.* Warming effects on the flux of CH₄ from peatland mesocosms are regulated by plant species composition: Richness and functional types. *Science of the Total Environment* **806**, 150831 (2022).
25. Kleine T, *et al.* Acclimation in plants - the Green Hub consortium. *Plant Journal* **106**, 23-

- 40 (2021).
26. Demmig-Adams B, Dumlao MR, Herzenach MK, Adams WW. Acclimation. In: *Encyclopedia of Ecology* (eds Jørgensen SE, Fath BD). Academic Press (2008).
 27. Bao T, Jia G, Xu X. Warming enhances dominance of vascular plants over cryptogams across northern wetlands. *Global Change Biology* **28**, 4097-4109 (2022).

Reviewers' Comments:

Reviewer #1:

Remarks to the Author:

I appreciate the authors' efforts in addressing my previous concerns. w. Yet, I have one concern left. The authors replace satellite NDVI/EVI with VPM-GPP, GOSIF GPP, FLUXCOM GPP. However, these GPP products, except for GOSIF GPP, are produced with the input of VPD, right? If so, these two GPP products are "inherently" biased by VPD, and direct satellite observations of NDVI/EVI and GOSIF GPP are better.

Other comments:

- In the abstract, the authors wrote "suggesting an indispensable role of northern peatlands in stabilizing the climate system." Remove it. VPD is only one environmental factor. Peatlands could be a carbon source due to other drivers.
- Suggest moving Figure 5 to supporting info.
- I strongly suggest authors improve the language with the help from an English-native speaker or English-editing service provider.
- Typo in Eq.4?

Overall, this is an important paper. I recommend publication after revisions.

Reviewer #2:

Remarks to the Author:

The authors have considered both reviewers' comments in their manuscript. With the removal of ESM results, the paper now focuses on experimental research and is much easier to follow. The manuscript is now improved, and I do not have other comments. I congratulate the authors.

REVIEWERS' COMMENTS

Reviewer #1 (Remarks to the Author):

I appreciate the authors' efforts in addressing my previous concerns. w. Yet, I have one concern left. The authors replace satellite NDVI/EVI with VPM-GPP, GOSIF GPP, FLUXCOM GPP. However, these GPP products, except for GOSIF GPP, are produced with the input of VPD, right? If so, these two GPP products are “inherently” biased by VPD, and direct satellite observations of NDVI/EVI and GOSIF GPP are better.

Reply: We appreciate the reviewer for the positive evaluation of our manuscript. As pointed out by the reviewer, in the latest version of the manuscript, we used VPM GPP, GOSIF GPP, and FLUXCOM GPP to replace NDVI and LAI to quantify the VPD impacts on vegetation growth. We also collected the eddy-covariance flux dataset, including FLUXNET2015 (global non-peatland regions, 95 sites) and FLUXNET-CH₄ Community Product (northern peatlands, 18 sites), to verify the robustness of our satellite-derived VPD effects. Both field- and grid-scale observations that investigated VPD impact based on GPP consistently showed the neutral VPD effects in the northern peatlands.

In response to the comment on the possible bias for using VPD in developing VPM-GPP and FLUXCOM GPP datasets, we double-checked our methodology and confirmed that VPD was not used in the production of VPM GPP and FLUXCOM GPP (Fig. 1 and Table 1). Enhanced Vegetation Index, Land Surface Water Index (LSWI), temperature, and photosynthetically active radiation were major variables for the VPM model¹ (Fig. 1). The LSWI uses the shortwave infrared and the near-infrared regions of the electromagnetic spectrum². Producing FLUXCOM GPP (RS+METEO) mainly used Normalized Difference Vegetation Index, Normalized Difference Water Index, Radiation, temperature, and Water Availability Index (WAI)^{3,4} (Table 1). WAI was estimated using a simple soil water balance model where VPD was not an input variable⁵. Taken together, VPD is not used, directly or indirectly, to produce VPM GPP and FLUXCOM GPP. Furthermore, VPM GPP⁶ and FLUXCOM GPP^{7,8} have been used to estimate the response of GPP to VPD at regional and global scales, which partially confirmed its feasibility in evaluating VPD impacts on vegetation. The above two lines of evidence support our analysis in the latest version of the manuscript. Three satellite-derived GPP datasets showed a consistent GPP response to VPD, as evidenced by correlation coefficients among $PCOR_{GOSIF\ GPP\ vs.\ VPD}$ (Partial correlation coefficients between GOSIF GPP and VPD), $PCOR_{VPM\ GPP\ vs.\ VPD}$, and $PCOR_{FLUXCOM\ GPP\ vs.\ VPD}$, all greater than 0.8 (Fig. 2).

Fig. 1 Schematic illustration of the methodology for producing VPM GPP¹. MODIS: Moderate Resolution Imaging Spectroradiometer; NCEP: National Centers for Environmental Prediction; EVI: enhanced vegetation index; LSWI: land surface water index; LST: land surface temperature; LUT: look-up table; T_{day}: daytime air temperature; T_{night}: nighttime land surface temperature; ε₀: maximum light use efficiency; T_{opt}: optimal temperature for photosynthesis; T_{max}: maximum temperature for photosynthesis; T_{min}: minimum temperature for photosynthesis; LSWI_{max}: maximum LSWI during the growing season. T_{scalar}: temperature limitation for photosynthesis; PAR: photosynthetically active radiation; W_{scalar}: water limitation for photosynthesis; fPAR_{chl}: fraction of PAR absorbed by chlorophyll; APAR_{chl}: absorbed PAR by chlorophyll.

		RS	RS + METEO
Product specifications	Spatial resolution	0.0833°	0.5°
	Temporal resolution	8 daily	daily
	Time period	2001–2015	Depending on climate forcing
	Climate input	n.a.	CRUNCEPv8, WFDEI, GSWP3, CERES-GPCP
	Tiling by PFT	no	yes
	Spatial & Seasonal patterns	f(RS)	f(RS, METEO)
	Interannual & trend patterns	f(RS)	f(METEO)
Training specifications	Machine learning methods	9: RF, ANN, GMDH, MARS, MTE (3 variants), KRR, SVR	3: RF, ANN, MARS
	Number of flux observations for training	~20,000	~200,000
	Spatial features	PFT, Max of MSC(fAPAR*Rg), Min of MSC(Rg)	PFT, Max of MSC(WA _{LJ}), Mean of MSC(BAND 6), Max of MSC(fAPAR*Rg)
	Spatial, seasonal features	Rpot, MSC(EVI*LST _{Day})	Rpot, MSC(NDWI), MSC(LST _{Night}), MSC(EVI*Rg)
	Spatial, seasonal, interannual features	Rg, LST _{Day} , Anom of LST _{Night} , Anom of (EVI*LST _{Day})	Rg, Rain, Rh, Rg*IWA*MSC(NDVI)

Table 1. List of acronyms^{3, 4}: Enhanced Vegetation Index (EVI), fraction of Absorbed Photosynthetically Active Radiation (fAPAR), daytime Land Surface Temperature (LST_{Day}) and

night time Land Surface Temperature (LST_{Night}), Normalized Difference Vegetation Index (NDVI), Normalized Difference Water Index (NDWI), Plant Functional Type (PFT), incoming global Radiation (R_g), top of atmosphere potential Radiation (R_{pot}), Index of Water Availability (IWA), Relative humidity (Rh), upper Water Availability Index WAI (WAIU), Mean Seasonal Cycle (MSC). Random forest (RF), Artificial Neural Network (ANN), Multivariate Adaptive Regression Splines (MARS), Model-Tree Ensemble (MTE), Kernel Ridge Regression (KRR), and Support Vector Regression (SVR).

Fig. 2 Correlations between any two of the three satellite-derived $PCOR_{GPP vs. VPD}$ (FLUXCOM GPP, VPM GPP, and GOSIF GPP). “r” indicates the correlation coefficients. GOSIF GPP vs. VPD, VPM GPP vs. VPD, FLUXCOM GPP vs. VPD indicate the partial correlation coefficients of GPP with VPD.

Other comments:

- **In the abstract, the authors wrote” suggesting an indispensable role of northern peatlands in stabilizing the climate system.” Remove it. VPD is only one environmental factor. Peatlands could be a carbon source due to other drivers.**

Reply: Thanks for your suggestions. Following the reviewer’s suggestions, we rewrote this sentence. The neutral VPD impacts in northern peatlands contrast with the reported vegetation suppression under rising VPD caused by concurrent warming and decreasing relative humidity in the global non-peatlands, suggesting a critical need in model improvement for representing VPD impacts in northern peatlands.

- **Suggest moving Figure 5 to supporting info.**

Reply: Thanks for your suggestions. Yet, the author team discussed this comment and decided to keep it in the main body of the text. Figure 5 is a conceptual diagram summarizing the major findings of this study; it is visual, concise, and comprehensive. Therefore, we prefer to keep it in the main text as it is.

- **I strongly suggest authors improve the language with the help from an English-native speaker or English-editing service provider.**

Reply: Thank you for pointing this out. Following the reviewer's suggestions, we have used an English editing company, which has edited the manuscript thoroughly. We read

throughout the manuscript to make final decisions on accepting or declining the edits by the company. The new manuscript is much improved regarding readability and clarity.

- Typo in Eq.4?

Reply: Thanks for pointing this out. We have revised the equation (4). Flux tower-based G_c (mm s^{-1}) was calculated by rearranging the Penman-Monteith equation using the following formula (Equation 4 and 5)^{9, 10}.

$$G_c = \left[\left(\frac{\Delta}{\gamma} \times \frac{H}{LE} - 1 \right) \times \gamma_a + \frac{\rho C_p}{\gamma} \times \frac{VPD}{LE} \right]^{-1} \quad (4)$$

Where Δ is the ratio of the change in SVP to T_a (Pa K^{-1}); γ is the psychometric constant (Pa K^{-1}); ρ is the air density (kg m^{-3}); C_p is the specific heat of air at constant pressure ($\text{J kg}^{-1} \text{K}^{-1}$); Following to a previous study¹¹ in equation 4, γ_a is the Aerodynamic resistance (s m^{-1} ,); k is the von Kármán constant ($k = 0.4$); ws is the wind speed.

Overall, this is an important paper. I recommend publication after revisions.

Reply: We thank the reviewer for supporting this paper and recognizing our effort in this revision.

Reviewer #2 (Remarks to the Author):

The authors have considered both reviewers' comments in their manuscript. With the removal of ESM results, the paper now focuses on experimental research and is much easier to follow. The manuscript is now improved, and I do not have other comments. I congratulate the authors.

Reply: We thank the reviewer for supporting our study and appreciate the constructive comments in the review process.

References

1. Zhang Y, *et al.* A global moderate resolution dataset of gross primary production of vegetation for 2000-2016. *Scientific Data* **4**, 170165 (2017).
2. Chandrasekar K, Sessa Sai MVR, Roy PS, Dwevedi RS. Land Surface Water Index (LSWI) response to rainfall and NDVI using the MODIS Vegetation Index product. *International Journal of Remote Sensing* **31**, 3987-4005 (2010).
3. Jung M, *et al.* The FLUXCOM ensemble of global land-atmosphere energy fluxes. *Sci Data* **6**, 74 (2019).
4. Jung M, *et al.* Scaling carbon fluxes from eddy covariance sites to globe: synthesis and evaluation of the FLUXCOM approach. *Biogeosciences* **17**, 1343-1365 (2020).
5. Tramontana G, *et al.* Predicting carbon dioxide and energy fluxes across global FLUXNET sites with regression algorithms. *Biogeosciences* **13**, 4291-4313 (2016).
6. Song Y, Jiao W, Wang J, Wang L. Increased Global Vegetation Productivity Despite Rising Atmospheric Dryness Over the Last Two Decades. *Earth's Future* **10**, e2021EF002634 (2022).
7. Zhong Z, *et al.* Disentangling the effects of vapor pressure deficit on northern terrestrial vegetation productivity. *Science Advances* **9**, eadf3166 (2023).
8. He B, *et al.* Worldwide impacts of atmospheric vapor pressure deficit on the interannual variability of terrestrial carbon sinks. *National Science Review* **9**, nwab150 (2022).
9. Dengel S, Grace J. Carbon dioxide exchange and canopy conductance of two coniferous forests under various sky conditions. *Oecologia* **164**, 797-808 (2010).
10. Blanken P, Black TA. The canopy conductance of a boreal aspen forest, Prince Albert National Park, Canada. *Hydrological Processes* **18**, 1561-1578 (2004).
11. Fu Z, *et al.* Atmospheric dryness reduces photosynthesis along a large range of soil water deficits. *Nature Communications* **13**, 989 (2022).